# Brief communication: An Ice surface melt scheme including the diurnal cycle of solar radiation

Uta Krebs-Kanzow[1], Paul Gierz[1], and Gerrit Lohmann[1]

[1]Alfred Wegener Institute Helmholtz Centre for Polar and Marine Research, Bremerhaven, Germany

**Correspondence:** Uta Krebs-Kanzow (uta.krebs-kanzow@awi.de)

**Abstract.**

We propose a surface melt scheme for glaciated land surfaces, which only requires monthly mean short wave radiation and temperature as inputs, yet implicitly accounts for the diurnal cycle of short wave radiation. The scheme is deduced from the energy balance of a daily melt period which is defined by a minimum solar elevation angle. The scheme yields a better spatial representation of melting than common empirical schemes when applied to the Greenland Ice Sheet, using a 1948-2016 regional climate and snow pack simulation as a reference. The scheme is physically constrained and can be adapted to other regions or time periods.

## 1 Introduction

The surface melt of ice sheets, ice caps and glaciers results in a freshwater runoff that represents an important freshwater source and directly influences the sea level on centennial to glacial-interglacial time scales. Surface melt rates can be determined from direct local measurements (e.g. Ahlstrom et al., 2008; Falk et al., 2018). On a larger scale, melt rates can be separated from integral observations such as the the World Glacier Monitoring Service (WGMS) (Zemp et al., 2015, and references therein) or the mass changes of ice sheets detected by the Gravity Recovery and Climate Experiment (GRACE) (Tapley et al., 2004; Wouters et al., 2014), which requires additional information about other components of the mass balance, such as basal melting, accumulation, sublimation and refreezing (Sasgen et al., 2012; Tedesco and Fettweis, 2012). In principal, the surface melt rate can be deduced from the net heat flux into the surface layer, as soon as the ice surface has been warmed to the melting point. For low solar elevation angles, however, the net heat flux into the surface layer usually becomes negative, the ice surface cools below the melting point and melting ceases. Consequently, energy balance modelling provides reliable surface melt rates only if sub-daily changes in ice surface temperature and nocturnal freezing are taken into account. Where sub-daily energy balance modelling is not feasible, surface melt is often estimated from empirical schemes. A common approach is the positive degree-day method as formulated e.g. in Reeh (1989). This particularly simple aproach linearly relates mean melt rates to positive degree-days, $PDD$, in which $PDD$ refers to the temporal integral of near surface temperatures T exceeding the melting point. The PDD-scheme is computationally inexpensive and requires only seasonal or monthly near surface air temperatures as input. Consequently, it has been applied in the context of long climate simulations (e.g. Charbit et al., 2013; Ziemen et al., 2014; Heinemann et al., 2014; Roche et al., 2014; Gierz et al., 2015) and paleo-temperature reconstructions (e.g. Box, 2013;

Wilton et al., 2017). Another empirical approach uses a linear function of solar radiation and temperature to predict surface melt. This approach was originally used to estimate ablation rates of glacial ice sheets (Pollard, 1980; Pollard et al., 1980). Formally similar schemes have been chosen, when the influence of solar radiation is changing over orbital time scales (the insolation temperature melt (ITM) equation designed to be used with monthly or seasonal forcing on long time scales, e.g. van den Berg et al., 2008; Robinson et al., 2010; de Boer et al., 2013), or for debris-covered glaciers, where surface albedo, and thereby the effect of insolation, is partly independent of air temperature (enhanced temperature index models (ETIM), consider sub-daily radiation and temperature of that period of a day, during which temperature exceeds a threshold temperature. Typically, this aproach is used with sub-daily climate forcing from weather stations, e.g. Pellicciotti et al., 2005; Carenzo et al., 2016). The empirical schemes, however, incorporate parameters, which require a local calibration and which are not necessarily valid under different climate conditions. Additionally, Bauer and Ganopolski (2017) demonstrate that the PDD-scheme fails to drive glacial-interglacial ice volume changes as it cannot account for albedo feedbacks. An alternative approach could be, to modify and simplify energy balance models in a way that reduces their data requirements and computational costs. Krapp et al. (2017) have formulated a complete surface mass balance model including accumulation, surface melt and refreezing (SEMIC) which can be used with daily or monthly forcing. SEMIC predicts the surface mass balance with a daily time step, but implicitly accounts for the sub-daily temperature variability in the surface layer of the ice to account for diurnal freeze-melt cycles.

In the following, we deduce a more simplified scheme from the energy balance, which is formally similar to the ETIM and ITM-schemes but incorporates physically constrained parameters. This new scheme only requires monthly means of temperature and solar radiation as input but implicitly resolves the diurnal cycle of radiation. In a first application on the Greenland Ice Sheet, GrIS, we use a simulation of Greenland's climate of the years 1948 to 2016 with the state-of-the-art regional climate and snow pack model MAR (version 3.5.2 forced with reanalysis data from the National Centers for Environmental Prediction–National Center for Atmospheric Research (NCEP) for the years 1948-2016, Kalnay et al., 1996; Fettweis et al., 2017) as a reference.

## 2   The daily melt period and its energy balance

The temperature of a surface layer of ice $T_i$ must rise to the melting point $T_0$ before the net energy uptake $Q$ of a surface layer can result in a positive surface melt rate $M$. In the following, we define background melt conditions on a monthly scale and melt periods on a daily scale.

The near surface air temperature $T_a$ usually does not exceed $T_0$ if (after winter) the ice is still too cold to aproach $T_0$ during daytime, so that, on a monthly scale surface air temperatures $\overline{T}_a$ (with the bar denoting monthly means hereafter) can serve as an indicator for background melting conditions. In the following we assume that monthly mean melt rates $\overline{M} > 0$ only occur if $\overline{T}_a > T_{min}$, where $T_{min}$ is a typical threshold temperature to allow melt.

The daily melt period shall be that part of a day, during which $T_i = T_0$ and $Q \geq 0$. Here, this period is assumed to be centered around solar noon, so that it is also defined by the period $\Delta t_\Phi$, during which the sun is above a certain elevation angle $\Phi$ (this minimum elevation angle will be estimated at the end of this section). Further, $q_\Phi$ is the ratio between the short wave radiation at the surface averaged over the daily melt period, $SW_\Phi$, and the short wave radiation at the surface averaged over the whole

day, $SW_0$, as

$$q_\Phi = \frac{SW_\Phi}{SW_0} \qquad (1)$$

Both $\Delta t_\Phi$ and $q_\Phi$ depend on the diurnal cycle of short wave radiation and can be expressed as functions of latitude and time for any elevation angle $\Phi$, if we include parameters of the Earth's orbit around the sun. $\Delta t_\Phi$ and $q_\Phi$ will be derived in Sect. 2.1. During the melt period, $Q_\Phi$ provides energy for fusion and results in a melt rate, which, averaged over a full day $\Delta t$, amounts to

$$M = \frac{Q_\Phi \Delta t_\Phi}{\Delta t \rho L_f} \qquad (2)$$

with latent heat of fusion $L_f = 3.34 \times 10^5\,\mathrm{J\,kg^{-1}}$ and the density of liquid water $\rho = 1000\,\mathrm{kg\,m^{-3}}$. The energy uptake of the surface layer is

$$Q_\Phi = (1-A)SW_\Phi + \epsilon_i LW\downarrow - LW\uparrow + R \qquad (3)$$

with surface albedo A, long wave emissivity of ice $\epsilon_i = 0.95$, downward and upward longwave radiation $LW\downarrow$ and $LW\uparrow$ respectively and the sum of all non radiative heat fluxes R. By definition,

$$LW\uparrow = \epsilon_i \sigma T_0^4 \qquad (4)$$

is valid during the melting period, with $\sigma = 5.67 \times 10^{-8}\,\mathrm{W\,m^{-2}\,K^{-4}}$ being the Stefan–Boltzmann constant. Further $T_a - T_0$ will be small relative to $T_0$ so that $LW\downarrow$ can be linearized to

$$LW\downarrow = \epsilon_a \sigma T_a^4 \approx \epsilon_a \sigma (T_0^4 + 4T_0^3(T_a - T_0)) \qquad (5)$$

with $\epsilon_a = 0.76$ being the emissivity of the near-surface air layer, if we neglect long wave radiation from upper atmospheric layers. Neglecting latent heat fluxes and heat fluxes to the subsurface and assuming R to be dominated by the turbulent sensible heat flux, we parameterize $R = \beta(T_a - T_0)$, with the coefficient $\beta$ representing the temperature sensitivity of the sensible heat flux. The coefficient $\beta$ primarily is a function of wind speed $u$ and according to Braithwaite (2009) can be estimated as $\beta = \alpha u$ with $\alpha \approx 4\,\mathrm{W\,s\,m^{-3}\,K^{-1}}$ at low altitudes. To find a formulation that is based on monthly climate forcing we need to estimate the mean melt period temperature from monthly mean temperatures. Near surface air temperature measurements from PROMICE stations on the GrIS reveal a good agreement between monthly mean temperatures of the daily melt periods and the $PDD_{\sigma=3.5}$ approximated as in Braithwaite (1985) from monthly mean near surface temperature $\overline{T}_a$ and a constant standard deviation of $\sigma = 3.5\,°\mathrm{C}$ (Fig. S1 in the supplement). Rewriting Eq. (3) for monthly means, we thus replace $(T_a - T_0)$ with $PDD_{\sigma=3.5}(\overline{T}_a)$. The above approximations and assumptions then yield an implicitly diurnal Energy Balance Model (dEBM), which only requires monthly mean temperatures and solar radiation as atmospheric forcing, while albedo may be parameterized as in common surface mass balance schemes (e.g. Krapp et al., 2017):

$$\overline{M} \approx \left(q_\Phi(1-A)\overline{SW}_0 + c_1 PDD_{\sigma=3.5}(\overline{T}_a) + c_2\right)\frac{\Delta t_\Phi}{\Delta t \rho L_f} \qquad (6)$$

where

$$
\begin{aligned}
c_1 &= \epsilon_i \epsilon_a \sigma 4 T_0^3 + \beta \\
&= 3.5\,\mathrm{W\,m^{-2}\,K^{-1}} + \beta \\
c_2 &= -\epsilon_i \sigma T_0^4 + \epsilon_a \epsilon_i \sigma (T_0^4) \\
&= -71.9\,\mathrm{W\,m^{-2}}
\end{aligned}
\tag{7}
$$

for any month that complies with the background melting condition $\overline{T}_a > T_{min}$. The sensitivity of the scheme to the choices of $\beta$ and to enhanced long wave radiation due to cloud cover or changed atmospheric composition is considered in sect. 4.

Both $q_\Phi$ and $\Delta t_\Phi$ strongly depend on latitude and month of the year. Thus, a given combination of insolation and temperature forcing yields different melt rates at different locations or seasons. The sensitivity of the dEBM to latitude is further investigated in sect. 4.

Finally, we use that $M = 0$ in the moment when the sun passes $\Phi$ and formulate the instantanous energy balance anlogously to Eq. (6) as

$$(1 - A)\tau \widehat{S}_r \sin\Phi + c_1 (T_a(\Phi) - T_0) + c_2 = 0. \tag{8}$$

with $\tau$ representing the transmissivity of the atmosphere over the melting surface, $\widehat{S}_0$ being the solar flux density at the top of the atmosphere (TOA), and the instantaneous air temperature $T_a(\Phi)$. The transmissivity $\tau$ strongly depends on cloud cover while $\widehat{S}_0$ only weakly varies seasonally due to the eccentricity of the orbit of the Earth. Assuming that $T_a(\Phi) \approx T_0$ and using one estimate of $\tau \widehat{S}_r$ for the melt season of the entire model domain, we can estimate

$$\Phi = \arcsin \frac{-c_2}{(1 - A)\tau \widehat{S}_r} \tag{9}$$

independent of time or location. The dEBM's sensitivity to the range of possible elevation angles is discussed in sect. 4.

## 2.1   Derivation of $\Delta t_\Phi$ and $q_\Phi$

The derivation of $\Delta t_\Phi$ and $q_\Phi$ is based on spherical trigonometry and fundamental astronomic considerations which, for instance, are discussed in detail in Liou (2002). The elevation angle $\vartheta$ of the sun changes throughout a day according to

$$\sin\vartheta = \sin\phi \sin\delta + \cos\phi \cos\delta \cos h(\vartheta) \tag{10}$$

with the latitude $\phi$, the solar inclination angle $\delta$ and the hour angle h. The time which the sun spends above an elevation angle $\vartheta$ then is

$$\Delta t_\vartheta = \frac{\Delta t}{\pi} h(\vartheta) = \frac{\Delta t}{\pi} \arccos \frac{sin\vartheta - \sin\phi \sin\delta}{\cos\phi \cos\delta}. \tag{11}$$

We assume that surface solar radiation is proportional to the TOA radiation $\widehat{S}_r$ throughout a day (i.e. we neglect that transmis-
sivity of the atmosphere $\tau$ is usually increasing with elevation angle and assume that cloud cover does not exhibit a diurnal

cycle). The solar radiation during the period which the sun spends above a certain elevation angle $\vartheta$ is then

$$SW_\vartheta = \frac{\tau \widehat{S_r}}{\pi \Delta t_\vartheta} \left( h(\vartheta) \sin\phi \sin\delta + (\cos\phi \cos\delta \sin h(\vartheta)) \right) \tag{12}$$

Eq. 12 also allows to estimate $\tau \widehat{S_r}$ from $SW_0$. Furthermore we can calculate the ratio between the mean short wave radiation during the melt period $SW_\Phi$ and the mean daily downward short wave radiation $SW_0$ at the surface independent of $\tau \widehat{S_r}$:

$$q_\Phi = \frac{SW_\Phi}{SW_0} = \frac{h(\Phi) \sin\phi \sin\delta + \cos\phi \cos\delta \sin h(\Phi)}{h(0) \sin\phi \sin\delta + \cos\phi \cos\delta \sin h(0)} \frac{\Delta t}{\Delta t_\Phi}. \tag{13}$$

## 3   First evaluation of the scheme

The dEBM and two empirical schemes are calibrated and evaluated using the state-of-the-art regional climate and snow pack model MAR (Fettweis et al., 2017) as a reference.

The elevation angle used in the dEBM is estimated as $\Phi = 17.5°$, aplying Eq. (9) with a typical albedo of 0.7 and $\tau \widehat{S_r} = 800\,\mathrm{W\,m^{-2}}$ being roughly estimated from the summer insolation in the ablation regions (Eq. 12). This estimate corresponds to a transmissivity of $\tau \approx 0.6$ which is in good agreement with Ettema et al. (2010). Further, the dEBM is optimized to reproduce the total annual Greenland surface melt averaged over the entire MAR-simulation by calibrating the background melting condition as $\overline{T}_a > -6.5\,°\mathrm{C}$ and the parameter $\beta = 10\,\mathrm{W\,m^{-2}\,K^{-1}}$. We then apply the scheme to $\overline{SW}_0$, $PDD_{\sigma=3.5}(\overline{T}_a)$ and albedo $A$ from a MAR-simulation of Greenland's climate (years 1948 to 2016) (Fettweis et al., 2017) and compare estimated melt rates with the respective MAR melt rates.

Two empirical schemes are considered in the same way: a PDD-scheme based on $PDD_{\sigma=5}(\overline{T}_a)$, as defined and calibrated in Krebs-Kanzow et al. (2018) and a scheme, in the following refered to as $dEBM_{const}$, which is a simplified variant of the dEBM where parameters are constant in time and space:

$$M = \left( (1-A)\overline{SW}_0 + k_1 PDD_{\sigma=3.5}(\overline{T}_a) + k_2 \right) \frac{1}{\rho L_f} \tag{14}$$

with $k_1 = 10\,\mathrm{W\,m^{-2}\,K^{-1}}$ and $k_2 = -55\,\mathrm{W\,m^{-2}}$. The $dEBM_{const}$ is very similar to the ITM-scheme and also uses similar parameters as in Robinson et al. (2010), but includes PDD instead of temperature, which particularly yields different results for low temperatures. As in Robinson et al. (2010), we treat $k_2$ as a tuning parameter to optimize the scheme and also use $\overline{T}_a > -6.5\,°\mathrm{C}$ as a background melting condition.

The computational cost of the dEBM in this application is very similar to the other two schemes as parameters are computed only once prior to the application. All schemes reproduce the total annual Greenland surface melt averaged over the entire MAR-simulation of $489\,\mathrm{Gt}$ with a relative bias not exceeding $1\,\%$ (the mean bias is $0.4\,\mathrm{Gt}$ for the PDD scheme, $-0.6\,\mathrm{Gt}$ for the $dEBM_{const}$ and $-2.0\,\mathrm{Gt}$ for the dEBM). These calibrations are primarily conducted to facilitate a fair comparison between the different schemes and are not necessarily optimal for other applications.

Equations (6) and (14) appear formally similar, with the second term being temperature dependent (the "temperature contribution") and the first and third term being independent of temperature and only depending on solar radiation (the "radiative

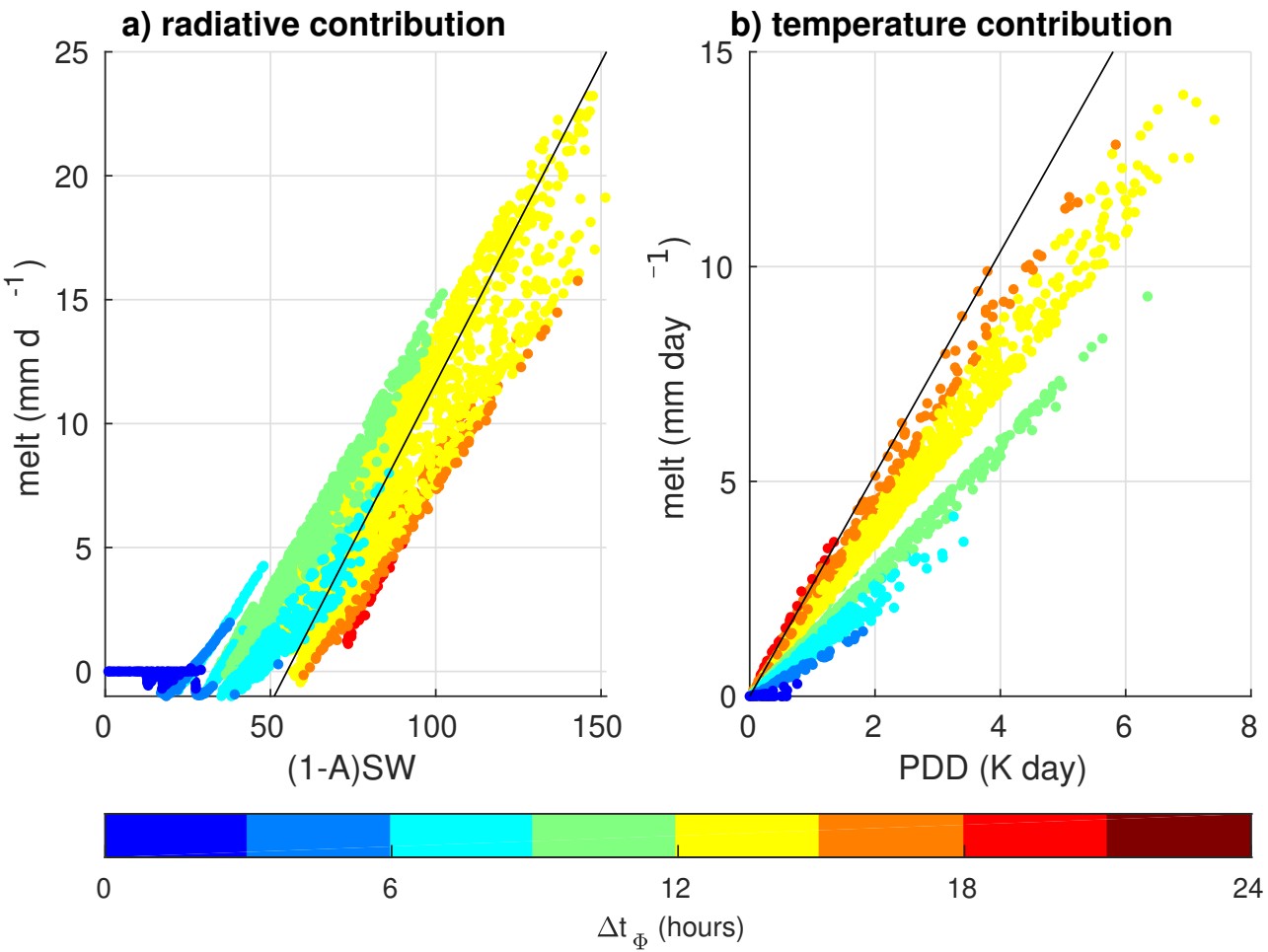

**Figure 1.** a) Contribution of the first and third term (radiative contribution) and b) of the second term (temperature contribution) in Eq. (6) to monthly melt rates as diagnosed with climatological temperatures and solar radiation from the MAR simulation. Colors indicate length of melt period (h). The black lines represent the respective prediction of the $dEBM_{const}$ according to Eq. (14)

contribution"). However, the respective parameters cannot be compared directly, as the $\Delta t_\Phi$ and $q_\Phi$ depend on latitude and month. $\Delta t_\Phi$ and $q_\Phi$ modulate the radiative contribution and $\Delta t_\Phi$ modulates the temperature contribution in Eq. (6). Fig. 1a illustrates the radiative and Fig. 1b the temperature contributions as diagnosed from the MAR simulation in comparison to the respective contribution from the $dEBM_{const}$. On the GrIS the radiative contribution can exceed $25\,\mathrm{mm\,d^{-1}}$ in the summer months and the two schemes appear qualitatively similar. The radiative contribution in the dEBM becomes less efficient for long melt periods, as the same insolation must balance the outgoing longwave radiation for a longer time. On the other hand, radiative contribution can also decrease towards short melt periods, if the sun only marginally rises above the minimum elevation angle at solar noon. In high latitudes, this effect becomes important for higher estimates of the minimum elevation angles (Sect. 4). The temperature contribution of the dEBM does not exceed $15\,\mathrm{mm\,d^{-1}}$ (Fig. 1b) and becomes more efficient with longer melt periods and would agree with the $dEBM_{const}$ for a melt period of 18 hours.

Atmospheric forcing (insolation and temperature) and albedo are here obtained from MAR output, and are fully consistent with the MAR melt rates. Consequently, we can evaluate the skill of the considered schemes independent of the quality of the atmospheric forcing and the representation of albedo. On the other hand, we can not evaluate the performance of the schemes for defective input. With respect to error propagation the PDD-scheme might be more robust and , as it only requires temperature as a forcing and only distinguishes between snow and ice but does not require albedo. Given the ideal input, all schemes reproduce the year-to-year evolution of the total Greenland surface melt of the MAR-simulation reasonably well (Fig. S3 in the supplement). The PDD-scheme yields increasing errors with intensifying surface melt rates, which is not apparent for the $dEBM_{const}$ and dEBM (Fig. 2). On the other hand, $dEBM_{const}$ particularly overestimates (underestimates) melt rates for very short (long) melt periods. In comparison to the two empirical schemes, the dEBM produces smaller local errors with biases being pronounced only in a narrow band along the ice sheet's margins (Fig.3).

## 4 Sensitivity to model parameters and boundary conditions

**Sensitivity to tuning parameters:** In the above application, the parameter $\beta$ for sensible heat and the background melting condition $T_{min}$ have served as tuning parameters. The parameter $\beta = 10\,\mathrm{W\,m^{-2}\,K^{-1}}$ was detemined by optimizing the scheme to MAR melt rates. This value agrees reasonably well with the moderate wind speeds found in PROMICE observations during melt periods (Fig. S2 in the supplement). Changing $\beta$ by $\pm 20\%$ changes the total annual Greenland surface melt by $\pm 3\%$. The choice of $T_{min} = -6.5\,^\circ\mathrm{C}$ is in good agreement with observations, which reveal no substantial melt for temperatures $< -7\,^\circ\mathrm{C}$ (e.g. Orvig, 1954). Increasing the background melting condition $T_{min}$ particularly reduces the melt rates at high elevations, while reducing $T_{min}$ results in a longer melting season and increases the annual surface melt. Using no background melting condition at all, results in unrealistic melt rates at high elevations and would almost double the predicted total Greenland surface melt. Changing $T_{min}$ by $\pm 1\,\mathrm{K}$ changes the predicted mean annual surface melt by $\pm 8\%$ for the MAR simulation used in this study. Intense surface melt is usually accompanied by warm temperatures and is thus insensitive to the choice of $T_{min}$. As refreezing particularly suppresses the contribution of weak surface melt at low temperatures, the resulting runoff can be expected to be less sensitive to the choice of $T_{min}$.

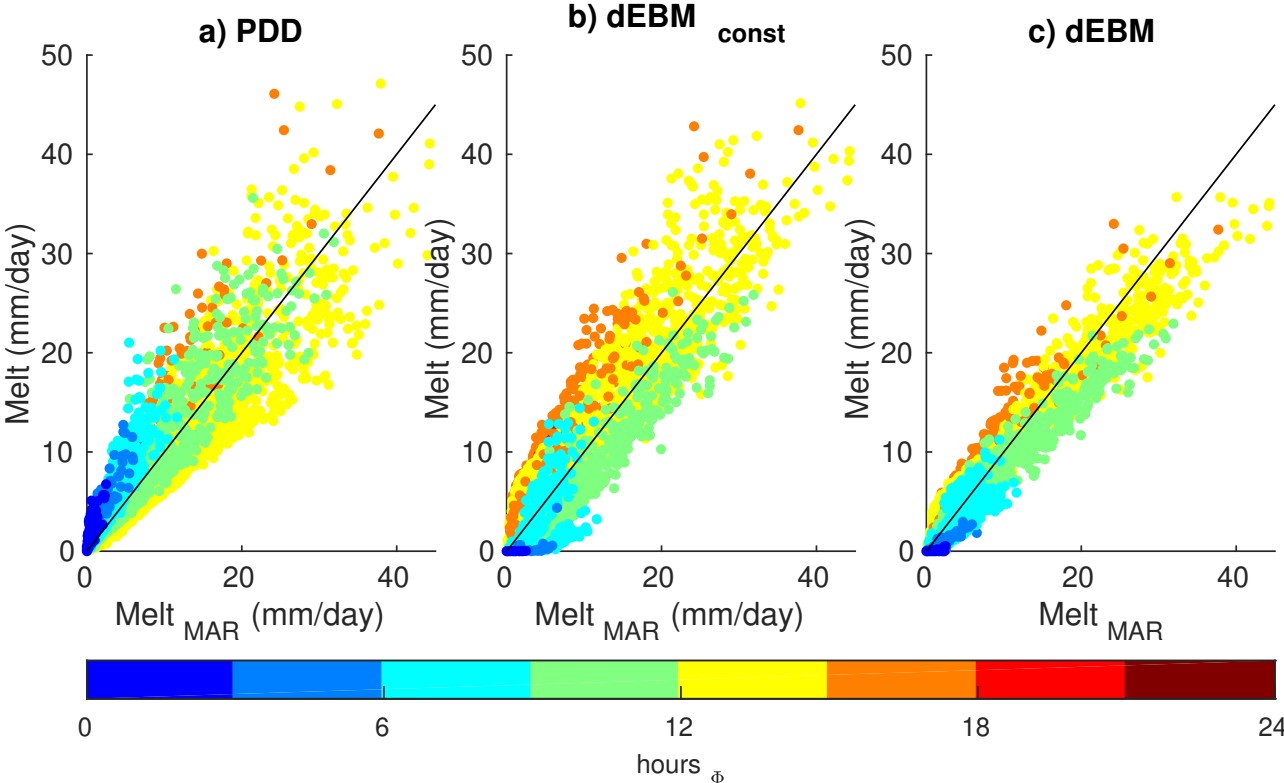

**Figure 2.** Multi-year monthly mean meltrates averaged of the years 1948-2016 as predicted by a) the PDD-scheme, b) the $dEBM_{const}$ and c) the dEBM against respective MAR melt rates. Colors reflect the length of the daily melt period. Identity is displayed as a black line in all panels for comparison.

**Sensitivity to diurnal cycle of solar radiation:** Melt schemes which do not include the diurnal cycle of radiation will predict the same melt rate for a given combination of insolation and temperature forcing, irrespective of latitude or season. By contrast, Fig. 4 indicates a strong sensitivity of the dEBM surface melt predictions to latitude in summer. According to the dEBM, a short melt period with intensive solar radiation is causing melt more effectively than a longer melt period with accordingly

5  weaker solar radiation. This sensitivity is particulary prominent in high latitudes and may explain the latitudinal bias found in many studies which do not resolve radiation on sub-daily time scales (e.g. Plach et al., 2018; Krebs-Kanzow et al., 2018; Krapp et al., 2017).

**Sensitivity to orbital configuration and transmissivity of the atmosphere:** The TOA solar flux density $\widehat{S_r}$ only depends on the distance between Earth and Sun and due to the eccentricity of the Earth's orbit gradually varies by $\pm3.5\%$ from the

10  solar constant from December to July respectively. On orbital time scales this seasonal deviation from the solar constant may amount to $10\%$. Transmissivity $\tau$ and emissivity $\epsilon_a$, on the other hand, strongly depend on cloud cover and atmospheric composition and additionally depend on the solar elevation angle. In consequence the minimum elevation angle $\Phi$ may be less

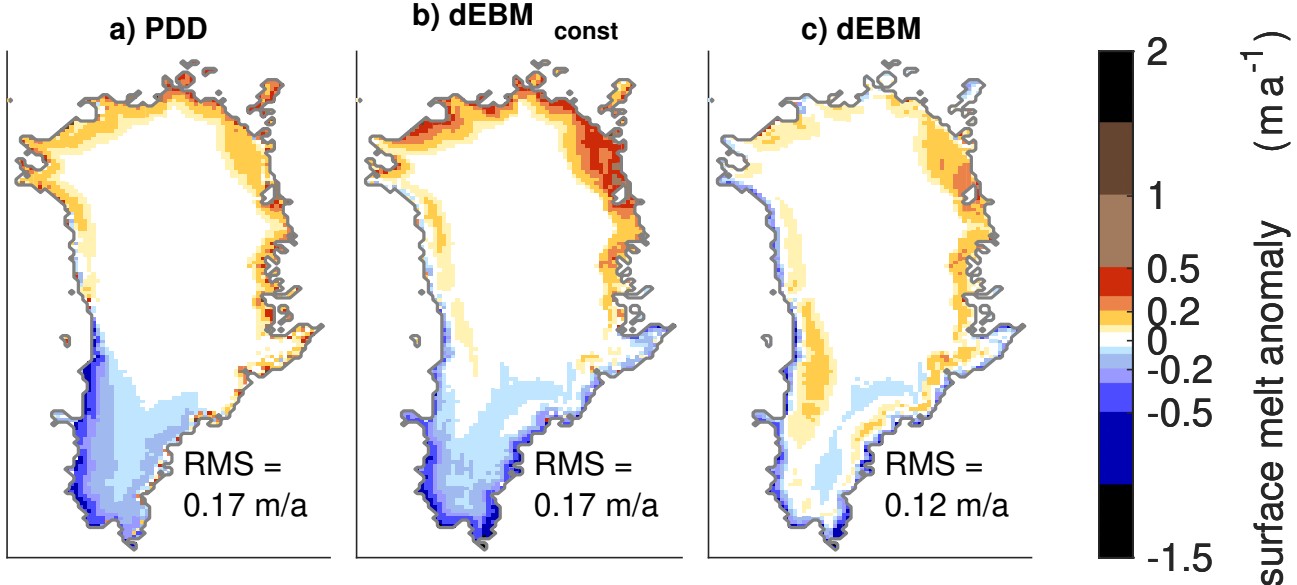

**Figure 3.** Bias between yearly melt rates as predicted by the individual schemes and as simulated by MAR, averaged over the whole simulation: a) PDD b) $dEBM_{const}$ c) the proposed new scheme dEBM. The respective root mean square error (RMS) is given in the individual panels.

then 13° ($\tau\widehat{S_r} = 1150\,\mathrm{W\,m^{-2}}$ for clear sky, intense summer insolation). For overcast sky and weak summer insolation, we can ultimately expect $\tau\widehat{S_r} < 400\,\mathrm{W\,m^{-2}}$. In that case, however, it is not justified to use the clear sky emissivity in Eqs. 5 and 7. Consequently, the proposed scheme is no longer suitable, as net outgoing long wave radiation will vanish and the energy balance will become very sensitive to turbulent heat fluxes. Applications aiming at continental ice sheets with climatological

forcing will be however restricted to a much narrower range of scenarios. As one can expect that transmissivity decreases towards the morning and afternoon hours, it may be justified to reduce the estimate of $\tau\widehat{S_r}$ by a few percent. Fig. 4 reveals that the scheme becomes very sensitive if the minimum elevation angle $\Phi$ takes values close to or larger than the obliquity of the Earth. Under such conditions the duration of the melt period will vanish near the Pole. On the other hand the scheme is remarkably insensitive to intensified insolation (and accordingly reduced elevation angle $\Phi$) or variations in the obliquity.

Accordingly, estimating the elevation angle locally and for each month using Eq. 12, which is possible but computationally more expensive, does not improve the skill of the dEBM noticiably (not shown).

## 5 Discussion and conclusion

The presented new scheme for surface melt (dEBM) requires, like the insolation temperature melt scheme (ITM), monthly mean air temperatures and insolation as input, but implicitly also includes the diurnal cycle. Together with suitable schemes

for albedo and refreezing (e.g. the parameterizations presented in Robinson et al., 2010), it may replace empirical surface melt

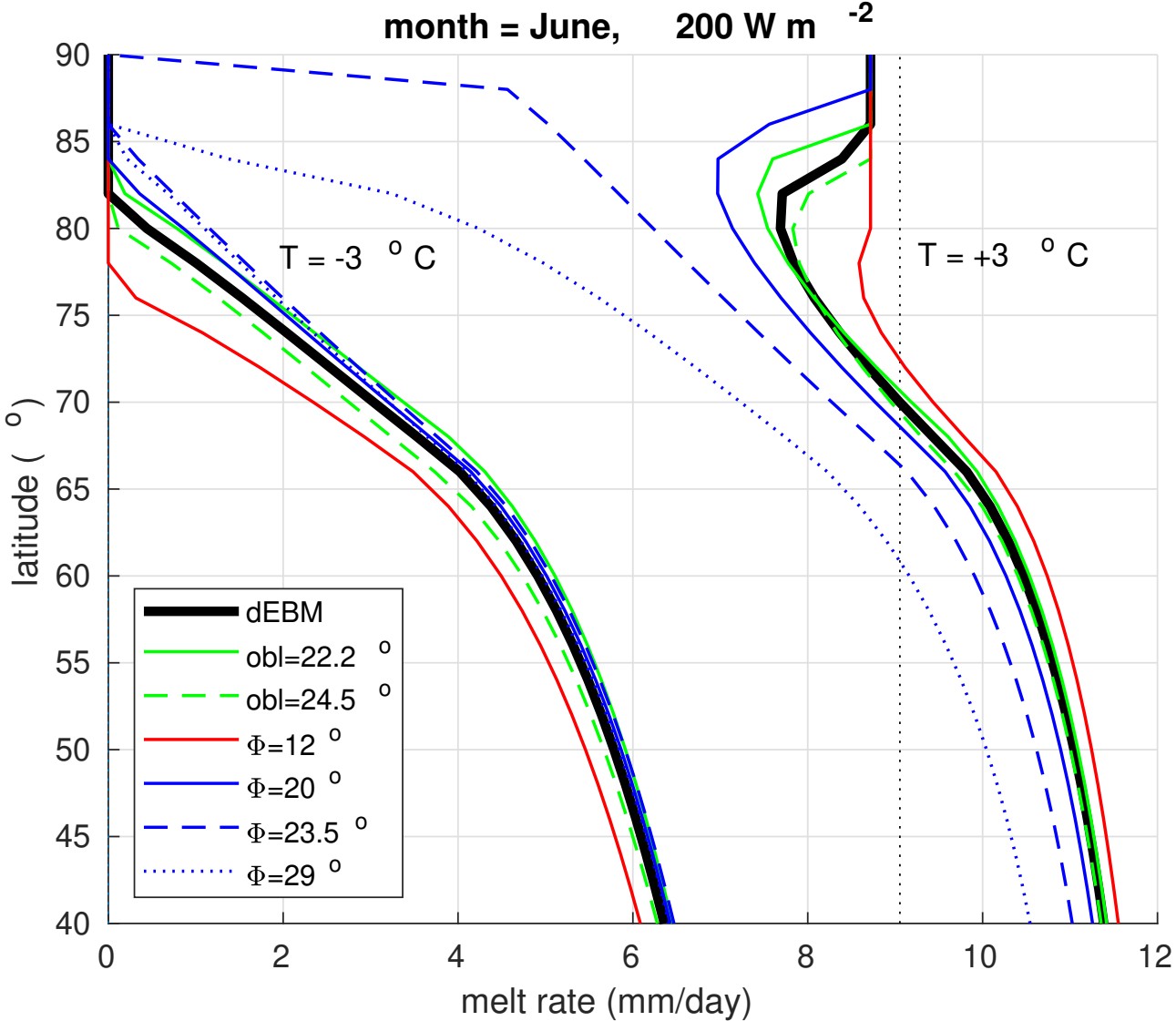

**Figure 4.** Sensitivity of the dEBM: June surface melt rate as predicted for $SW_0 = 200\,\mathrm{W\,m^{-2}}$, A = 0.7, $T_a = -3\,°\mathrm{C}$ (left curves) and $T_a = 3\,°\mathrm{C}$ (right curves). Black: predictions with parameters as used for the presented simulation of Greenland's surface melt. Green: parameters are recalculated using the minimum (solid) and maximum (dashed) obliquity of the last 1 million years. Blue: parameters are recalculated for minimum elevation angles $\Phi$ of 20° (solid), 23.5° (dashed) and 29° (dots) corresponding to a reduced solar density flux at the surface of $\tau\widehat{S}_r = 700\,\mathrm{W\,m^{-2}}$, $\tau\widehat{S}_r = 600\,\mathrm{W\,m^{-2}}$ and $\tau\widehat{S}_r = 500\,\mathrm{W\,m^{-2}}$, respectively. Red: parameters are recalculated for $\Phi = 12°$, which corresponds to an intensified solar density flux at the surface of $\tau\widehat{S}_r = 1150\,\mathrm{W\,m^{-2}}$. The $dEBM_{const}$ predicts 0 mm/day for $SW_0 = 200\,\mathrm{W\,m^{-2}}$, A = 0.7, $T_a = -3\,°\mathrm{C}$ and 9 mm/day for $SW_0 = 200\,\mathrm{W\,m^{-2}}$, A = 0.7, $T_a = 3\,°\mathrm{C}$ (black dots).

schemes which are commonly used in ice sheet modelling on long time scales.

An application to the Greenland Ice Sheet indicates, that the scheme may improve the spatial representation of surface melt in comparison to common empirical schemes. However, an evaluation to an independent data base is desirable. The most important advantage of the dEBM over empirical schemes may be, that it can be globally applied to other ice sheets and glaciers and under different climate conditions, as parameters in the scheme are physically constrained and implicitly account for the orbital configuration.

In the presented formulation a threshold temperature serves as a prerequisite for surface melt on monthly time scales. This threshold temperature should be considered as a tuning parameter, as the representation of the ice-atmosphere boundary layer in Earth system models may differ considerably from the MAR simulation, which here has served as a reference. Furthermore, long wave radiation and non-radiative heat fluxes are only crudely represented. Depending on the application, it may be advisable to adapt the parameterization of turbulent heat fluxes and long wave radiation to different climate regimes in order to account for changed wind speed, humidity, cloud cover or greenhouse gas concentration.

The daily melt period is defined by a minimum solar elevation angle. Together with the melt period, parameters in the dEBM depend on latitude and month of the year, but do not change from year to year if the minimum solar elevation angle is kept constant and the orbital configuration remains the same. For the Greenland Ice Sheet, a minimum solar elevation angle of $17.5°$ was roughly estimated from the mean summer insolation normal to a surface at the bottom of the atmosphere. The dEBM is very sensitive if the intensity of solar radiation is substantially weaker than in the presented application (e.g. due to cloud cover or atmospheric water content). In this case it is necessary to carefully re-estimate the minimum elevation angle and to adjust the model parameters accordingly. Otherwise, the scheme appears to be relatively insensitive to changes in the orbital configuration and the parameters choices in this study may be valid in a wider range of settings.

The presented formulation has been designed for long Earth System Model applications, but it may be adapted to be also used in the context of climate reconstructions or to be applied on regional or local scales. Furthermore, having defined the daily melt period by the minimum elevation angle, it should also be possible to estimate the amount of refreezing by considering the energy balance of the remainder of the day, following a similar approach as in Krapp et al. (2017).

*Code availability.* A matlab version of the dEBM is available under https://github.com/ukrebska/dEBM/

*Competing interests.* The authors declare that they have no competing interests

*Acknowledgements.* We would like to thank Xavier Fettweis for providing MAR model output. Further we are grateful for the valuable comments and constructive suggestions from Alexander Robinson, Mario Krapp and an anonymous referee. U. Krebs-Kanzow is funded by the Helmholtz Climate Initiative REKLIM (Regional Climate Change), a joint research project of the Helmholtz Association of German research

centres. P. Gierz is funded by the German Ministry of Education and Research (BMBF) German Climate Modeling Initiative PalMod. This work is part of the project "Global sea level change since the Mid Holocene: Background trends and climate-ice sheet feedbacks" funded from the Deutsche Forschungsgemeinschaft (DFG) as part of the Special Priority Program (SPP)-1889 "Regional Sea Level Change and Society" (SeaLevel).

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
