# Peer review of "Brief communication: An Ice surface melt scheme including the diurnal cycle of solar radiation"

_The Cryosphere, 2018_

## Referee Comment (RC1) · A. Robinson (Referee) · 7 Aug 2018

This manuscript presents an alternative approach to calculating surface melt on an ice sheet that is more complex than the PDD method, but still simpler than the intermediate complexity alternatives available so far. It is a creative approach, refreshing and a very welcome addition to a relatively small set of melt models available for long time-scale ice sheet modeling. It indeed seems to present significant benefits over other methods. Therefore I would very much like to see this work published in TC.

That said, I find the manuscript in need of major revisions (see major and minor comments below). I simply get the feeling that the manuscript is not complete. I would

prefer to see a slightly longer paper with at least some sensitivity analysis (How sensitive is the presented model to the parameter choices? How important is the diurnal aspect of the model?) and deeper discussion of subtleties of the approach (e.g., T_min, PDD(T_a) versus T_a – see comments below), and its comparison to other methods.

With such revisions, I would then prefer to see this as a normal article rather than a "Brief Communication" (although I understand this is an editorial decision).

– Major comments ————

The introduction does a nice job of concisely laying out the problem. However, the description of the alternative methods is not very precise, and I think the comparison with them could be more thorough and analytical.

- As pointed out by the editor, a distinction should be made when discussing a melt model alone (which is a somewhat artificial construct in isolation) versus an energy balance model, which may calculate many variables that are useful for ice-sheet modeling (ice temperature, albedo, refreezing, smb). Thus I see PDD, ETIMs, ITM (see next point) and the model presented here in a similar category – melt models that can be used as a subcomponent of smb models – while SEMIC and full EBMs are more wholistic solutions. A bit of clarity here on these definitions would improve the manuscript greatly.

-"ETIM" as used throughout the text seems to be the wrong term for the comparison being made here. ETIMs involve a "temperature index" such as PDD (Hock et al., 2003; Pelliccioti et al., 2005). This is why, in Robinson et al., 2010, we opted to create the name ITM for the model of Eq. 14, since we saw it simply as an "insolation-temperature melt model".

- Equation 14 is not correct, in representing the approach of Pollard (1980) or Robinson et al. (2010). The second term should not contain PDDs, but the mean near-surface air temperature relative to the melting point: k_1*(T_a-T_0).

[Figure]

- SEMIC also supports the input of monthly temperature data, although the model itself is calculated on daily time steps (from Krapp et al., 2017: "In principle, the use of monthly input data is also supported but would require interpolation to daily time steps."). I would additionally note that SEMIC is open source and prepared to run easily with MAR data as input, making its comparison with dEBM feasible if the authors wanted to be more thorough. It would certainly be convincing if it could be explicitly shown that dEBM can do a better job than SEMIC for a much lower cost. (This point is only a suggestion, and I would not consider it necessary for revision.)

I find the approach outlined here quite elegant and the physical derivation is nicely described. However, then I am surprised to see PDD pop up in Eq. 6 again. Would it not be simpler keep $(T\_a-T\_0)$ here? The only reason to use PDDs is to incorporate a measure of variability in Ta. But it seems to me that if you want to include the variability around $T\_a$ in the melt model, it would be more appropriate to apply it to the whole equation rather than just to the temperature term (ie, calculate the average melt rate from the distribution of melt rates for the month).

If you follow the path above, this change would make Eq. 6 essentially equivalent to Eq. 14 (also without PDDs), and it maintains its physically-based origins and makes it obvious that the key differences are: - The term $q\_\Phi$, which scales the insolation according to the time it is actually available. - The term $dt\_\Phi$, which scales the melt according to the time when it is relevant. - The derivation of the constants $c\_1$ and $c\_2$.

I note that the used values of $c\_1 = 14.4$ and $c\_2 = -71.9$ are not too far from values used in Eq. 14 for $k\_1 = 10$ and $k\_2 = -60.5$. It would be interesting to understand if this is systematic, that generally $c\_1 > k\_1$ and $k\_2 > c\_2$, to compensate for the lack of $q\_\Phi$ and $dt\_\Phi$ terms. For example, if in Eq. 6, you set $q\_\Phi = 1$ and $dt\_\Phi = dt$, how well does your model perform (after retuning the constants) – as well as before, or is the performance degraded? In other words, I would be happy to see an analysis that specifically shows the value of incorporating the diurnal terms to the model.

– Minor comments –––

Units and variables: Please check the units carefully. For example, $T_a$ is in Kelvin, but then $T_{min} = (T_0-6.5)$ K, right? Also, in Eq. 14, is the first term "$SW_0$" the same as "$SWD_0$" defined earlier in the text? Please keep the same terms throughout.

Page 1, line 14: information on => information about

Page 1, line 15: refreeze => refreezing

Page 1, line 23: computational => computationally

Page 1, line 24: temperatures. => temperatures as input.

Page 1, line 25: or paleo-temperature => and paleo-temperature

Page 2, line 1: aproach => approach

Page 2, line 1: "Another empirical aproach, the enhanced temperature-index method, ETIM" <= In addition to the fact that I believe ETIM is the wrong term here, as I already mentioned, ETIM refers to a class of models that can take many forms that generally extend PDD in various ways, not to a specific model formulation. Therefore, I would rephrase here. Alternatively, you can use the term "ITM", which does refer to the formulation of Pollard (1980). Or, a more descriptive term for this model would be "linearized EBM" (Pollard, 1980).

Page 2, line 19: a surface melt rate => a non-zero/positive surface melt rate

Page 3, line 5 (Eq. 3): I see no reason why $e_i$ should appear multiplied with $LW_{down}$. This is only relevant for $LW_{up}$ (as in Eq. 4), correct?

Page 3, line 7: Per definitionem => By definition

Page 3, line 23 (Eq. 7): It looks like $c_1$ is missing the term $e_a$, following the current equation formulations.

Page 4, Section 2.1: Please make sure to use the same variables and notation as in

the rest of the text. I guess that the elevation angle $\Phi$ in the previous section is the same as the elevation angle $\theta$ in Sect. 2.1.

Page 4, Eq. 13: I would suggest adding the intermediate definition of q_$\Phi$ here to remind readers of your previous definition: q_$\Phi$ = SW_$\Phi$ / SW_0 = [full definition]. Also again be clear about SW versus SWD.

Page 4, line 23: What is the calculation of $\Phi$ = 23.5° used for later? As I understand all tests were using MAR albedo, etc. Is this just an example?

Page 5, line 11: Eqations => Equations

Page 5, line 11: "Eqations (6) and (14) appear formally similar, with the first and third term representing the radiative contribution and the second term representing the PDD contribution." <= This sentence is contaminated by the mistake in Eq. 14, however, just thinking about it in terms of Eq. 6, it is clear from the derivation that the first term represents shortwave radiation and the second and third terms represent the net longwave radiation and heat fluxes from R combined. Please rephrase.

Page 5, line 11-21: Generally, I find this paragraph difficult to follow. Is the "flat elliptic" referring to the orbital configuration of the Earth, or some pattern in the figure itself? Does "going along with" mean "causing"? I find that "PDD contribution" a not very convenient name for the second term in Eq. 6, since it is easily confused with the PDD melt model itself in this context. I would consider serious revision here for clarity.

Page 5, line 23: derived => obtained?

Page 5, line 25: "defective input" <= I'm not quite sure what you want to say with this sentence, consider rephrasing somewhat. Wouldn't it be possible to make your ideal input data "defective" for testing purposes, if that was your goal?

Page 5, line 27: due to => Given

Page 5, line 32: It does not seem appropriate to limit the comparison of dEBM to points

that satisfy T_a > -6.5 C. Either the value of T_min=-6.5 C is adequate, or T_min should be set to a lower value. In either case, the correct choice of this threshold should be reflected by the comparison to MAR melt. Based on the horizontal line of dark blue points in Figs. 1 & 2, I have to guess that the threshold chosen here is too high, or for some reason the dEBM underestimates melt at low temperatures. This should be discussed in the paper clearly.

Page 7, line 1: biasses => biases

Page 7, Figure 2 caption: lenght => length

Page 7, line 6: refreeze => refreezing

Page 7, line 6: used together with the enhanced temperature index method in => presented by

Page 8, line 13: "This threshold temperature should be considered as a tuning parameter" <= I had understood this T_min simply to be a cost-saving measure, to avoid calculating the melt model for points where melt would be zero. However, this sentence makes me believe that the parameter is more important than I realized. Please elaborate on the role of T_min more in the derivation section for clarity.

Page 8, line 16: Depending on application => Depending on the application

– References –––

Hock et al., Journal of Hydrology, doi:10.1016/S0022-1694(03)00257-9, 2003

Pellicciotti et al., Journal of Glaciology, doi:10.3189/172756505781829124, 2005

---

## Referee Comment (RC2) · Anonymous Referee #2 · 15 Aug 2018

This manuscript presents a novel surface melt scheme for glaciated land surfaces that accounts for the diurnal cycle of short wave radiation; a diurnal Energy Balance Model (dEBM). The manuscript is intended as a Brief Communication which introduces and explains the new dEBM scheme and provides an initial evaluation with respect to two other empirical schemes and a state-of-the-art regional climate model.

I feel this works well as a Brief Communication that introduces the new scheme and provides a simple initial assessment. I find the mathematical derivation of the scheme to be generally well explained with references to other papers for further details. An initial assessment of the dEBM scheme is shown by calculating melt rates for the Green-

land Ice Sheet (1948 – 2016) for which dEBM generally performs better than two other empirical schemes with the authors recognising dEBM does not predict correct melt rates in one specific circumstance.

I have some minor alterations to suggest,

Page 1 line 23 "computational" should be "computationally"

Page 2 line 27 "Further, we define the ratio between . . ..." would "qΦ is the ratio between . . . " read better?

Page 2 line 27 & 28 "SW" is used for the mean solar radiation. I assume the S and W stand for Short Wave, so it would be better to state that here

Page 3 lines 24 & 26 and Page 4 line 10 These lines appear to have been indented / tabbed

Page 4 line 12 TOA is introduced for "top of the atmosphere" but only used once on line 15 then not used on line 16 (where there are hyphens between the words). Is an initialism really needed?

Page 4 line 21 "Choosing $\beta$ = 10 . . . ". If that is a choice, i.e. if alternative values could have been chosen, then the reason for this specific choice should be given, e.g. cited or explained. If however it is the only reasonable value then it's not a choice and "using "would be better than "choosing"

Page 4 line 22 "-6.5K" $^{\circ}$C not K

Page 4, bottom, section 3 Just a general comment that any further citations or justifications for the values of coefficients used that can be included would be useful

Page 5 line 11 Misspelling "eqations"

Page 5 line 15 "GrIS" is used without definition. Whilst it is a well known abbreviation, especially for this journal, it maybe better to define it.

Page 5 line 16 "going along with . . .. " would "corresponding to . . . " read better?

Page 5 line 30 This refers to the blue points in Fig 2 panel 3 at 0 on the y-axis. I think this should be stated in the text.

Page 8 line 16 This is a new paragraph, should it be?

Figure S1, caption "meltrates" should be 2 words. Misspelling "lenght". Refers to PDD, ETIM and DEBM as a), b) and c) but they are not labelled as such in the figures. Also "Identity . . . black line" is not shown (I think perhaps the caption is for an earlier version of the figure?)

In the introduction it is mentioned that the PDD scheme is computationally inexpensive (page 1 line 23) and that energy balance models could have their computational costs reduced (page 2 line 7) but the evaluation makes no mention of the computational costs of dEBM and the other schemes. I think it would be useful to include a brief comment on the relative computational costs in section 3.

---

## Referee Comment (RC3) · M. Krapp (Referee) · 15 Aug 2018

This paper presents a new melt scheme which can be applied over glaciated surfaces such as ice sheets or glaciers. Its novel component is the latitude-dependent diurnal cycle of solar radiation thereby making it flexible enough to be applied for different regions and over different periods of time. The paper introduces an innovative melt scheme to complement existing melt schemes such as PDD or ETIM models. The model accounts for time- and latitude-depending changes in the diurnal cycle of incoming solar radiation which makes it appealing and very relevant to the ice sheet modelling community. The paper is well written and and the derivation of the model

equations is elegant but the paper has a few shortcomings and therefore needs major revisions before I would recommend it for publication in TC.

I also agree with Alex (one of the other reviewers) that an article rather than a "brief communication" would be the better format for this paper.

Major comments

- Solar elevation angle and surface slope: Whereas large parts of the Greenland ice sheet are rather flat its margins, where most of the melt occurs) are not and glaciers are even more sensitive to the slope of the embedding terrain. I suspect that the daily solar elevation angle depends on how the ice surface faces the sun. How much of an effect would a surface slope have and could that be included in Sect. 2.1?

- I expect the atmospheric transmissivity (Sect. 2.1) to decrease with increasing solar zenith angle. How much of an effect would that have?

- I think that using a single parameter for the emissivity of air ($\epsilon_a$) is also too simplistic and the contribution of cloud cover is missing. $LW^\downarrow$ is parameterised using $\epsilon_a$, which is the clear sky emissivity but how do you deal with cloudy skies? In fact, $\epsilon_a$ can vary between 0.7 (clear sky) and 1.0 (fully overcast). Therefore, the value for $c_2$ can vary between -90 and 0 W/m2 if you account for varying $\epsilon_a$. That means that a full overcast sky would add about 90 W/m2 to the surface energy uptake $Q$.

- I think in Eq. (7), $\epsilon_a$ is missing in the term for $c_1$, i.e., $c_1 = \epsilon_a \epsilon_i \sigma 4 T_0^3 + \beta$. If that is the case $c_1$ also yields a different value in line 25 on page 4 and my above argument about varying $\epsilon_a$ implies that $c_1$ can vary 13 and 14.4 W/m2K.

- Sensitivity of model parameters: I agree with the other reviewer that this paper

benefits from a sensitivity analysis of the paper. First, because most of the parameters have been fixed for a "First evaluation of the scheme" (Sect. 3) and second, it helps the reader to see how the model responds to different assumed parameter choices. For example, is the choice of $\beta$, $T_{min}$, or $\epsilon_a$ arbitrary or representative of the Greenland ice sheet? What does the reader need to change to apply this model to Antarctica and/or other ice caps and glaciers? In the conclusion you state the dEBM "can be applied to other ice sheets and glaciers and under different climate conditions". I think statement can be underpinned by a thorough parameter sensitivity analysis and, perhaps, a recommendation for those different conditions (e.g., for the more recent or deeper past)

- The PDD component of dEBM is in general smaller than in ETIM (Fig. 1b). Obviously, the PDD contribution of dEBM would be larger for a larger $\beta$ which can range between 7 to 20 W/m2K as you said earlier.

- I would like to see a plot showing the time series of monthly melt and different diagnostics (as is shown in the supplement). For example, melt rates and its individual components (the PDD and the ETIM-related term) in Eq. (6), or the parameterised short- and longwave radiation $SW$ and $LW^{\downarrow}$ would help the reader to understand what the model is doing internally. Specifically it would be nice to see how $q_{\phi}$, which is the novel part of your melt scheme, changes over time.

- To me everything in the conclusion, except for the first paragraph, is more like a "summary and discussion" section than an actual conclusion. Please revise.

- I guess if you consider a revision as article you can easily move Figure S1 (which is the only item in the supplement) to the main text.

- Out of curiosity (not needed for the revison): If the melt scheme just uses a few input parameters, is it possible to force it with atmospheric data from available observations of the GrIS? For example, GC-MET (http://cires1.colorado.edu/steffen/

gcnet/) or PROMICE (https://www.promice.dk/home.html)

Minor comments

- p2 ll.27-29: It is not clear whether $SW_0$ or $SW_\phi$ mean surface or TOA shortwave radiation.

- p.5 l22: Please, specify what the atmospheric forcing variables from the MAR model are.

- Please add a table with model parameters and parameter values used in the main text and analysis.

- Fig. 2: add units to axis labels; duplicate y-axis labels ("PDD", "ETIM", and "dEBM")

- Fig. 3: the min/max colors are really dark and hard to see

---

## Author Comment (AC1) · 9 Oct 2018

[tc,manuscript]copernicus

[separate-uncertainty=true, multi-part-units=single]siunitx [normalem]ulem color

**1 General remarks**

We would like to thank all reviewers for the reading of the manuscript and for their thoughtful and very constructive comments. In particular addressing the sensitivity of

the model and questioning the individual parameter choices has led to some interesting results. In the following, reviewer comments are printed in blue and manuscript excerpts are printed in grey. A texdiff-file is also provided for the revised manuscript. Also, I provide all figures from the Supplement at the end of this document. I will answer all points individually further below , but before that I would like to introduce some modifications of the dEBM, which were a side product of the revision process.

1.) I realized, that the minimum solar elevation angle was based on an incorrect estimate of the solar flux density at the surface ($\tau \widehat{S}_r = 600\square$.) Upon closer inspection I could identify the error in the calculation and corrected this to $\tau \widehat{S}_r = 800\square$, which results in a smaller minimum elevation angle of $\Phi = 17.5^o$.

2.) Analysing hourly PROMICE data, I found that $PDD_{\sigma=3.5}$ better represents melt period temperatures than $PDD_{\sigma=5}$ (approximated with a constant standard deviation of $\sigma = 3.5$ instead of $\sigma = 5$).

All experiments and analyses have been repeated, using $PDD_{\sigma=3.5}$ and minimum elevation angle $\Phi = 17.5$. We found that these corrections don't change the results qualitatively.

Furthermore we have included a section on the sensitivity of the scheme in the main manuscript. Depending on the editors decision this may as well be moved to the supplement:

**2  Sensitivity to model parameters and boundary conditions**

**Sensitivity to tuning parameters:**   In the above application, the parameter $\beta$ for sensible heat and the background melting condition $T_{min}$ have served as tuning parameters. The parameter $\beta = 10\square$ was detemined by optimizing the scheme to MAR melt rates. This value agrees reasonably well with the moderate wind speeds found in PROMICE observations during melt periods (Fig. S2 in the supplement). Chang-

./BILDER/Fig4.jpg

ing $\beta$ by $\pm 20\%$ changes the total annual Greenland surface melt by $\pm 3\%$. The choice of $T_{min} = -6.5$ is in good agreement with observations, which reveal no substantial melt for temperatures $< -7$ (e.g. **?**). Increasing the background melting condition $T_{min}$ particularly reduces the melt rates at high elevations, while reducing $T_{min}$ results in a longer melting season and increases the annual surface melt. Using no background melting condition at all results in unrealistic melt rates at high elevations and would almost double the predicted total Greenland surface melt. Changing $T_{min}$ by $\pm 1$ changes the predicted mean annual surface melt by $\pm 8\%$ for the MAR simulation used in this study. Intense surface melt is usually accompanied by warm temperatures and is thus insensitive to the choice of $T_{min}$. As refreezing particularly suppresses the contribution of weak surface melt at low temperatures, the resulting runoff can be expected to be less sensitive to the choice of $T_{min}$.

**Sensitivity to diurnal cycle of solar radiation:** Melt schemes which do not include the diurnal cycle of radiation will predict the same melt rate for a given combination of insolation and temperature forcing, irrespective of latitude or season. By contrast, Fig. 1 indicates a strong sensitivity of the dEBM surface melt predictions to latitude in summer. According to the dEBM, a short melt period with intensive solar radiation is causing melt more effectively than a longer melt period with accordingly weaker solar radiation. This sensitivity is particularly prominent in high latitudes and may explain the latitudinal bias found in many studies which do not resolve radiation on sub-daily time scales (e.g. **???**).

**Sensitivity to orbital configuration and transmissivity of the atmosphere:** The TOA solar flux density $\widehat{S_r}$ depends on the distance between Earth and Sun and due to the eccentricity of the Earth's orbit gradually varies by $\pm 3.5\%$ from the solar constant from December to July respectively. On orbital time scales this seasonal deviation from the solar constant may amount to $10\%$. Transmissivity $\tau$, on the other hand, strongly depends on cloud cover and atmospheric composition and additionally increases with the solar elevation angle. In consequence the minimum elevation angle $\Phi$ may be less then 13 ($\tau \widehat{S_r} = 1150\square$ for clear sky, intense summer insolation). For overcast sky and

weak summer insolation, we can ultimately expect $\widehat{\tau S_r} < 400\square$. In that case, however, it is not justified to use the clear sky emissivity in Eqs. 5 and 7. Consequently, the proposed scheme is no longer suitable, as net outgoing long wave radiation will vanish and the energy balance will become very sensitive to turbulent heat fluxes. Applications aiming at continental ice sheets with climatological forcing will be however restricted to a much narrower range of scenarios. As one can expect that transmissivity decreases towards the morning and afternoon hours, it may be justified to reduce the estimate of $\widehat{\tau S_r}$ by a few percent. Fig. 1 reveals that the scheme becomes very sensitive if the minimum elevation angle $\Phi$ takes values close to or larger than the obliquity of the Earth. Under such conditions the duration of the melt period will vanish near the Pole. On the other hand the scheme is remarkably insensitive to intensified insolation (and accordingly reduced elevation angle $\Phi$) or variations in the obliquity. Accordingly, estimating the elevation angle locally and for each month using Eq. **??**, which is possible but computationally more expensive, does not improve the skill of the dEBM noticiably (not shown).

**3  Response to first referee (Alexander Robinson)**

– Major comments ————– ... However, the description of the alternative methods is not very precise, and I think the comparison with them could be more thorough and analytical.
We now distinguish between ETIM amd ITM:
Introduction: " Another empirical approach uses a linear function of solar radiation and temperature to predict surface melt. This approach was originally used to estimate ablation rates of glacial ice sheets (**??**). Formally similar schemes have been chosen, when the influence of solar radiation is changing over orbital time scales (the insolation temperature melt (ITM) equation designed to be used with monthly or seasonal forcing on long time scales, e.g. **???**), or for debris-covered glaciers, where surface albedo,

and thereby the effect of insolation, is partly independent of air temperature (enhanced temperature index models (ETIM), consider sub-daily radiation and temperature of that period of a day, during which temperature exceeds a threshold temperature. Typically, this aproach is used with sub-daily climate forcing from weather stations, e.g. **??**). The empirical schemes, however, incorporate parameters, which require a local calibration and which are not necessarily valid under different climate conditions.

- As pointed out by the editor, a distinction should be made when discussing a melt model alone (which is a somewhat artificial construct in isolation) versus an energy balance model, which may calculate many variables that are useful for ice-sheet modeling (ice temperature, albedo, refreezing, smb). Thus I see PDD, ETIMs, ITM (see next point) and the model presented here in a similar category – melt models that can be used as a subcomponent of smb models – while SEMIC and full EBMs are more wholistic solutions. A bit of clarity here on these definitions would improve the manuscript greatly.
We have included a more detailed description of SEMIC:

Introduction: **?** have formulated a complete surface mass balance model including accumulation, surface melt and refreezing (SEMIC) which can be used with daily or monthly forcing. SEMIC predicts the surface mass balance with a daily time step, but implicitly accounts for the sub-daily temperature variability in the surface layer of the ice to account for diurnal freeze-melt cycles.

-"ETIM" as used throughout the text seems to be the wrong term for the comparison being made here. ETIMs involve a "temperature index" such as PDD (Hock et al., 2003; Pelliccioti et al., 2005). This is why, in Robinson et al., 2010, we opted to create the name ITM for the model of Eq. 14, since we saw it simply as an "insolation-temperature melt model".
- Equation 14 is not correct, in representing the approach of Pollard (1980) or Robinson

et al. (2010). The second term should not contain PDDs, but the mean near-surface air temperature relative to the melting point: $k_1 * (T_a - T_0)$

Admittedly, this was formulated too sloppy. Trying to correctly incorporate the ITM concept, I noticed that this method has problems to reproduce melt rates at low temperatures. In the context of a full SMB-model this would propably carry no weight as refreeze is usually balancing the surface melt at low temperatures. However, the calibration to MAR melt rates is hampered by the biased low temperature melt rates. In consequence, our calibration would yield parameters, which fail to skillfully reproduce the surface melt rates at warm temperatures. I have decided to stick to a scheme which incorporates PDD, but we now refer to this scheme as $dEBM_{const}$. Comparing to a dEBM version with constant parameters also meets your suggestion further below. We have reformulated the first part of section

**First evaluation of the scheme:** The dEBM and two empirical schemes are calibrated and evaluated using the state-of-the-art regional climate and snow pack model MAR (**?**) as a reference.

The elevation angle used in the dEBM is estimated as $\Phi = 17.5$, aplying Eq. (9) with a typical albedo of 0.7 and $\tau \widehat{S}_r = 800\square$ being roughly estimated from the summer insolation in the ablation regions (Eq. **??**). This estimate corresponds to a transmissivity of $\tau \approx 0.6$ which is in good agreement with **?**. Further, the dEBM is optimized to reproduce the total annual Greenland surface melt averaged over the entire MAR-simulation by calibrating the background melting condition as $\overline{T}_a > -6.5$ and the parameter $\beta = 10\square$. We then apply the scheme to $\overline{SW}_0$, $PDD_{\sigma=3.5}(\overline{T}_a)$ and albedo $A$ from a MAR-simulation of Greenland's climate (years 1948 to 2016) (**?**) and compare estimated melt rates with the respective MAR melt rates.

Two empirical schemes are considered in the same way: a PDD-scheme based on $PDD_{\sigma=5}(\overline{T}_a)$, as defined and calibrated in **?** and a scheme, in the following refered to as $dEBM_{const}$, which is a simplified variant of the dEBM where parameters are constant in time and space:

$$M = ((1 - A)\overline{SW}_0 + k_1 PDD_{\sigma=3.5}(\overline{T}_a) + k_2)\frac{1}{\rho L_f} \qquad (1)$$

with $k_1 = 10\square$ and $k_2 = -55\square$. The $dEBM_{const}$ is very similar to the ITM-scheme and also uses similar parameters as in ?, but includes PDD instead of temperature, which particularly yields different results for low temperatures. As in ?, we treat $k_2$ as a tuning parameter to optimize the scheme and also use $\overline{T}_a > -6.5$ as a background melting condition.

The computational cost of the dEBM in this application is very similar to the other two schemes as parameters are computed only once prior to the application. All schemes reproduce the total annual Greenland surface melt averaged over the entire MAR-simulation of $489$ with a relative bias not exceeding $1$ (the mean bias is $0.4$ for the PDD scheme, $-0.6$ for the $dEBM_{const}$ and $-2.0$ for the dEBM). These calibrations are primarily conducted to facilitate a fair comparison between the different schemes and are not necessarily optimal for other applications.

- SEMIC also supports the input of monthly temperature data, although the model itself is calculated on daily time steps (from Krapp et al., 2017: "In principle, the use of monthly input data is also supported but would require interpolation to daily time steps."). I would additionally note that SEMIC is open source and prepared to run easily with MAR data as input, making its comparison with dEBM feasible if the authors wanted to be more thorough. It would certainly be convincing if it could be explicitly shown that dEBM can do a better job than SEMIC for a much lower cost. (This point is only a suggestion, and I would not consider it necessary for revision.)

I now discuss SEMIC more specificly in the Introduction (see above). I would also be very interested in this comparison. The intention behind the comparison of the different schemes in the presented paper, however, is not a complete intercomparison, but rather to demonstrate, that the latitudinal bias found in other schemes, might be

reduced, if we account for the diurnal cycle of radiation.

I find the approach outlined here quite elegant and the physical derivation is nicely described. However, then I am surprised to see PDD pop up in Eq. 6 again. Would it not be simpler keep $(T_a - T_0)$ here? The only reason to use PDDs is to incorporate a measure of variability in $T_a$. But it seems to me that if you want to include the variability around $T_a$ in the melt model, it would be more appropriate to apply it to the whole equation rather than just to the temperature term (ie, calculate the average melt rate from the distribution of melt rates for the month).

We have added a few lines to

**The daily melt period and its energy balance:** Near surface air temperature measurements from PROMICE stations on the GrIS reveal a good agreement between monthly mean temperatures of the daily melt periods and the $PDD_{\sigma=3.5}$ approximated as in ? from monthly mean near surface temperature $\overline{T}_a$ and a constant standard deviation of $\sigma = 3.5$ (Fig. S1 in the supplement).

And added the following to the

**Supplement: Mean surface temperture and wind speed of melt periods from observations** $T_a$ is the monthly mean temperature and thus also includes temperatures outside of the daily melt period. The strategy in our paper is to only consider that part of the day, when the ice is warm enough to melt. We thus need to estimate the mean temperature during this melt period. To illuminate the relation between $T_a$ and $T_{MP}$, we analyzed hourly climate data from PROMICE (?) weather stations: 2m air temperature $T_a$, surface temperature $T_{surf}$, albedo A and short wave radiation SW. In analogy to the dEBM, we determine the melt period for each month by identifying those hours which comply with the conditions

$$\overline{(1-A)SW} > 71.9 Wm^{-2}$$

and

$$\overline{T_{surf}} > -0.01^o C$$

**Fig. 2.** Monthly mean melt period temperature $T_{MP}$ and PDDs as functions of monthly mean near surface air temperature $T_a$. Crosses reflect monthly mean $T_{MP}$ as calculated from hourly near surface air temperature data of 18 PROMICE stations. Red and green points reflect PDD calculated from $T_a$ assuming a constant standard deviation of $3.5^oC$ and $5^oC$ respectively.

**Fig. 3.** Monthly mean wind speed during melt periods $u_{MP}$ as a function of monthly mean near surface air temperature $T_a$.

. The bars denotes hourly data taken from the monthly mean diurnal cycle. We analyzed 18 PROMICE stations which cover a period of up to ten years (2008-2017) and identified 390 monthly mean diurnal cycles which exhibit a melt period acording to our above definition. We don't need to resort to a minimum elevation angle here, as hourly radiation is available. Likewise the background melting condition is replaced by the condition, that hourly surface temperature data must be near melting point. Indeed, the PROMICE data indicate that PDD is quite a good proxy for the monthly mean temperature of the melt period $T_{MP}$. Using a constant standard deviation of $3.5$ exhibits a particularly good fit (Fig. S1).

Furthermore analyzing the mean wind speed during the above melt periods, we find on average a wind speed of $u_{MP} = 3.8$ (Fig. S2).

Why PDD is such a good estimate of $T_{MP}$ is not completely clear. It would be interesting to develop alternative estimates for $T_{MP}$.

Looking into the observational data, we also noticed that daily melt periods are considerably longer than in our original MAR-based study (up to 20 hours). In fact this is how I realized that the estimated minimum elevation angle must be incorrect. The newly estimated $\Phi = 17.5^o$ yields longer melt periods and generally agrees with the PROMICE data.

If you follow the path above, this change would make Eq. 6 essentially equivalent to Eq. 14 (also without PDDs), and it maintains its physically-based origins and makes it obvious that the key differences are: - The term $q_\Phi$, which scales the insolation according to the time it is actually available. - The term $dt_\Phi$, which scales the melt according to the time when it is relevant. - The derivation of the constants $c_1$ and $c_2$. I note that the used values of $c_1 = 13.5$ and $c_2 = -71.9$ are not too far from values used in Eq. 14 for $k_1 = 10$ and $k_2 = -60.5$. It would be interesting to understand if this is systematic, that generally $c_1 > k_1$ and $k_2 > c_2$, to compensate for the lack of $q_\Phi$ and $dt_\Phi$ terms. For example, if in Eq. 6, you set $q_\Phi$= 1 and $dt_\Phi$= dt, how well does your model perform (after retuning the constants) – as well as before, or is the performance degraded? In other words, I would be happy to see an analysis that specifically shows the value of incorporating the diurnal terms to the model

I think with introducing $dEBM_{const}$, this suggestion is implemented now. $dt_\Phi < dt$ in most places and accordingly $c_1 > k_1$. Understanding $q_\Phi$ is more complicated. We have reformulated the part about the radiative contribution and hope this is more clear, now. **First evaluation of the scheme:** The radiative contribution in the dEBM becomes less efficient for long melt periods, as the same insolation must balance the outgoing longwave radiation for a longer time. On the other hand, radiative contribution can also decrease towards short melt periods, if the sun only marginally rises above the minimum elevation angle at solar noon. This effect becomes important for higher estimates of the minimum elevation angles in high latitudes (Sect. 4).

**– Minor comments —–**

We would like to comment on the following minor comments
: Page 2, line 1: "Another empirical aproach, the enhanced temperature-index method, ETIM" <= In addition to the fact that I believe ETIM is the wrong term here, as I already mentioned, ETIM refers to a class of models that can take many forms that generally extend PDD in various ways, not to a specific model formulation. Therefore, I would

rephrase here. Alternatively, you can use the term "ITM", which does refer to the formu- lation of Pollard (1980). Or, a more descriptive term for this model would be "linearized EBM" (Pollard, 1980).

We now use $dEBM_{const}$

Page 3, line 5 (Eq. 3): I see no reason why $e_i$ should appear multiplied with $LW_{down}$. This is only relevant for $LW_{up}$ (as in Eq. 4), correct?

To my understanding, $e_i$ is also influencing, how much LW radiation is absorbed, as a good emitter is also a good absorber.

Page 4, line 23: What is the calculation of $\Phi = 23.5$ used for later? As I under-stand all tests were using MAR albedo, etc. Is this just an example?

Actually $\Phi$ is a crucial parameter, as $\delta t_\Phi$ and $q_\Pi$ will change with $\Phi$.

Page 5, line 11: "Eqations (6) and (14) appear formally similar, with the first and third term representing the radiative contribution and the second term representing the PDD contribution." <= This sentence is contaminated by the mistake in Eq. 14, however, just thinking about it in terms of Eq. 6, it is clear from the derivation that the first term represents shortwave radiation and the second and third terms represent the net longwave radiation and heat fluxes from R combined. Please rephrase.

We rephrased:

Equations (6) and (14) appear formally similar, with the second term being temper-ature dependent (the "temperature contribution") and the first and third term being independent of temperature and only depending on solar radiation (the "radiative contribution").

Page 5, line 11-21: Generally, I find this paragraph difficult to follow. Is the "flat elliptic" referring to the orbital configuration of the Earth, or some pattern in the figure

itself? Does "going along with" mean "causing"? I find that "PDD contribution" a not very convenient name for the second term in Eq. 6, since it is easily confused with the PDD melt model itself in this context. I would consider serious revision here for clarity.
We reformulated this paragraph (ecliptic was the wrong term):

Fig. 1a illustrates the radiative and Fig. 1b the temperature contributions as diagnosed from the MAR simulation in comparison to the respective contribution from the $dEBM_{const}$. On the GrIS the radiative contribution can exceed $25$ in the summer months and the two schemes appear qualitatively similar. The radiative contribution in the dEBM becomes less efficient for long melt periods, as the same insolation must balance the outgoing longwave radiation for a longer time. On the other hand, radiative contribution can also decrease towards short melt periods, if the sun only marginally rises above the minimum elevation angle at solar noon. This effect becomes important for higher estimates of the minimum elevation angles in high latitudes (Sect. 4). The temperature contribution of the dEBM does not exceed $15$ (Fig. 1b) and becomes more efficient with longer melt periods and would agree with the $dEBM_{const}$ for a melt period of 18 hours.

Page 5, line 25: "defective input" <= I'm not quite sure what you want to say with this sentence, consider rephrasing somewhat. Wouldn't it be possible to make your ideal input data "defective" for testing purposes, if that was your goal?
We slightly rephrased:

With respect to error propagation the PDD-scheme might be more robust and , as it only requires temperature as a forcing and only distinguishes between snow and ice but does not require albedo.

t does not seem appropriate to limit the comparison of dEBM to points that sat­isfy $T_a > -6.5C$. Either the value of $T_{min} = -6.5C$ is adequate, or $T_{min}$ should be set to a lower value. In either case, the correct choice of this threshold should be reflected by the comparison to MAR melt. Based on the horizontal line of dark blue points in

Figs. 1 and 2, I have to guess that the threshold chosen here is too high, or for some reason the dEBM underestimates melt at low temperatures. This should be discussed in the paper clearly. The horizontal line of dark blue points in Fig. 2 was related to the incorrect estimate of minimum elevation angle. We removed the RMS part from the main text and included the root mean square errors of the mean 1948-2016 local yearly surface melt rates in Fig. 3. We did not use the background melting condition for this calculation. The idea to use $T_{min}$ arose as I wanted to limit the analysis to the ablation region and melt season. I see now that this can bias the statistics (it did not substantially, though)

Page 8, line 13: "This threshold temperature should be considered as a tuning pa- rameter" <= I had understood this $T_{min}$ simply to be a cost-saving measure, to avoid calculating the melt model for points where melt would be zero. However, this sentence makes me believe that the parameter is more important than I realized. Please elaborate on the role of $T_{min}$ more in the derivation section for clarity
This is now discussed in the sensitivity section (Sect. 4).

We fully followed the following minor comments and corrected the manuscript accordingly:
Units and variables: Please check the units carefully. For example, $T_a$ is in Kelvin, but then $T_{min} = (T_0 - 6.5)K$, right? Also, in Eq. 14, is the first term "$SW_0$" the same as "$SWD_0$" defined earlier in the text? Please keep the same terms throughout
. Page 1, line 14: information on => information about
Page 1, line 15: refreeze => refreezing
Page 1, line 23: computational => computationally
Page 1, line 24: temperatures. => temperatures as input.
Page 1, line 25: or paleo-temperature => and paleo-temperature
Page 2, line 1: aproach => approach
Page 2, line 19: a surface melt rate => a non-zero/positive surface melt rate

Page 3, line 7: Per definitionem => By definition

Page 3, line 23 (Eq. 7): It looks like $c_1$ is missing the term $e_a$, following the current equation formulations.

Page 4, Section 2.1: Please make sure to use the same variables and notation as in the rest of the text. I guess that the elevation angle $\Phi$ in the previous section is the same as the elevation angle $\Theta$ in Sect. 2.1.

Page 4, Eq. 13: I would suggest adding the intermediate definition of $q_\Phi$ here to remind readers of your previous definition: $q_\Phi = SW_\Phi/SW_0$= [full definition]. Also again be clear about SW versus SWD.

Page 5, line 11: Eqations => Equations

Page 5, line 23: derived => obtained?

Page 5, line 27: due to => Given

Page 7, line 1: biasses => biases

Page 7, line 6: refreeze => refreezing

Page 7, line 6: used together with the enhanced temperature index method in =>presented by

Page 8, line 16: Depending on application => Depending on the application

**4 Response to second referee (anonymous)**

Page 4 line 12 TOA is introduced for "top of the atmosphere" but only used once on line 15 then not used on line 16 (where there are hyphens between the words). Is an initialism really needed?
It is now used more often

Page 4 line 21 "Choosing $\beta = 10$ ...". If that is a choice, i.e. if alternative values could have been chosen, then the reason for this specific choice should be given,

e.g. cited or explained. If however it is the only reasonable value then it's not a choice and "using "would be better than "choosing"

We now make clear that this is the outcome of an calibration: Further, the dEBM is optimized to reproduce the total annual Greenland surface melt averaged over the entire MAR-simulation by calibrating the background melting condition as $\overline{T}_a > -6.5$ and the parameter $\beta = 10\square$

Page 4, bottom, section 3 Just a general comment that any further citations or justifications for the values of coefficients used that can be included would be useful

We now reference **?** and **?** which agrees well with our independently calibrated parameters.

In the introduction it is mentioned that the PDD scheme is computationally inexpensive (page 1 line 23) and that energy balance models could have their computational costs educed (page 2 line 7) but the evaluation makes no mention of the computational costs of dEBM and the other schemes. I think it would be useful to include a brief comment on the relative computational costs in section 3.

We added the following line to section 3: The computational cost of the dEBM in this application is very similar to the other two schemes as parameters are computed only once prior to the application.

The following suggestions are obsolte after the modification of the manuscript: Page 5 line 16 "going along with .... " would "corresponding to..." read better?
Page 5 line 30 This refers to the blue points in Fig 2 panel 3 at 0 on the y-axis. I think this should be stated in the text.
Page 8 line 16 This is a new paragraph, should it be

[Figure]

We fully followed the following recommendations: Page 1 line 23 "computational" should be "computationally"
Page 2 line 27 "Further, we define the ratio between ....." would "$q_\Phi$ is the ratio between..." read better?
Page 2 line 27 and 28 "SW" is used for the mean solar radiation. I assume the S and W stand for Short Wave, so it would be better to state that here
Page 3 lines 24 and 26 and Page 4 line 10 These lines appear to have been indented /tabbed
Page 4 line 22 "-6.5K" $^o C$ not K

Page 5 line 11 Misspelling "eqations"
Page 5 line 15 "GrIS" is used without definition. Whilst it is a well known abbreviation, especially for this journal, it maybe better to define
Figure S1, caption "meltrates" should be 2 words. Misspelling "lenght". Refers to PDD, ETIM and DEBM as a), b) and c) but they are not labelled as such in the figures. Also "Identity... black line" is not shown (I think perhaps the caption is for an earlier version of the figure?)

**5  Response to third referee (Mario Krapp)**

**Major comments** - Solar elevation angle and surface slope: Whereas large parts of the Greenland ice sheet are rather flat its margins, where most of the melt occurs) are not and glaciers are even more sensitive to the slope of the embedding terrain. I suspect that the daily solar elevation angle depends on how the ice surface faces the sun. How much of an effect would a surface slope have and could that be included in

**Sect. 2.1?**

Indeed, the melt period may be extended/shortened by a southward/northward slope. If we would want to account for this, it would make it necessary to perform a projection of the solar radiation to the surface before estimating the minimum elevation angle locally. The slopes on the 20km grid of the MAR-simulation rarely exceed $1\%$, which could change the minimum elevation angles by $\approx 1^o$. Seeing that the manuscript is already quite lengthy, I decided to not include this into the manuscript.

- I expect the atmospheric transmissivity (Sect. 2.1) to decrease with increasing solar zenith angle. How much of an effect would that have?

This is a very good point, I would estimate that this effect may increase the elevation angle by up to $2^o$ relative to an estimate using a constant transmissivity. It is difficult to account for this effect in an objective way, but luckily the scheme is quite insensitive to minor changes in the minimum elevation angle and it appears sufficient to only do a rough estimate. I added the following sentence to the Sect. 4:

As one can expect, that transmissivity decreases towards the morning and afternoon hours, it may be justified to reduce the estimate of $\tau \widehat{S_r}$ by a few percent.

I think that using a single parameter for the emissivity of air ( $\epsilon_a$) is also too simplistic and the contribution of cloud cover is missing. $LW_{down}$ is parameterised using $\epsilon_a$ , which is the clear sky emissivity but how do you deal with cloudy skies? In fact, $\epsilon_a$ can vary between 0.7 (clear sky) and 1.0 (fully overcast). Therefore, the value for $c_2$ can vary between -90 and 0 W/m2 if you account for varying $\epsilon_a$. That means that a full overcast sky would add about 90 W/m2 to the surface energy uptake Q.

A very valid point, I did not consider this originally. The dEBM concept propably comes to its limits here. I have added this point to the sensitivity section as given above. However for continental ice sheets (i.e. Greenland and Antarctica and, in cold climates, the North American and Fennoscandian ice sheets) the clear sky assumption appears justified.

I think in Eq. (7), $\epsilon_a$ is missing in the term for $c_1$ , ... If that is the case $c_1$ also yields a different value in line 25 on page 4 and my above argument about varying $\epsilon_a$ implies that $c_1$ can vary 13 and 14.4 W/m2K
This is an error which was only in the text and not in my dEBM function. I corrected the text accordingly.

Sensitivity of model parameters:... We now include the section about sensitivity, as stated above

The PDD component of dEBM is in general smaller than in ETIM (Fig. 1b). Obviously, the PDD contribution of dEBM would be larger for a larger $\beta$ which can range between 7 to 20 W/m2K as you said earlier
This has changed after correcting and modifying various details. Also citing **?** we now provide better constraints for the choice of $\beta$.

I would like to see a plot showing the time series of monthly melt and different diagnostics (as is shown in the supplement). For example, melt rates and its individual components (the PDD and the ETIM-related term) in Eq. (6), or the parameterised short- and longwave radiation SW and $LW_{down}$ would help the reader to understand what the model is doing internally. Specifically it would be nice to see how $q_\Phi$ , which is the novel part of your melt scheme, changes over time.
Primarily, $q_\Phi$ and $\delta_\Phi$ affect surface melt latitudinally, and to some degree seasonally. Perhaps, the effects are sufficiently illustrated in the new Fig. 4. I am hesitant to add another figure on the seasonal effect, as the paper seems already quite long.

To me everything in the conclusion, except for the first paragraph, is more like a "summary and discussion" section than an actual conclusion. Please revise. We changed the title of the section accordingly. I guess if you consider a revision as article

you can easily move Figure S1 (which is the only item in the supplement) to the main text
This can be easily done, but I would leave this decision to the editor.

Out of curiosity (not needed for the revison): If the melt scheme just uses a few input parameters, is it possible to force it with atmospheric data from available observations of the GrIS? For example, GC-MET (http://cires1.colorado.edu/steffen/
At least the PROMICE data have a high frequency, so that better estimates should be possible, if a full energy balance model is used. Nevertheless, I can imagine that the scheme could be modified in a way, so that distributed melt estimates could be derived from satelite data in combination with weather station data. Also it could be possible to estimate melt rates from glaciers where weather stations only exist below the glacier. In both cases I would think that the scheme would have to undergo considerable modification. I would be indeed interested to discuss this with people from the observational community.

**Minor comments**

p2 ll.27-29: It is not clear whether $SW_0$ or $SW_\Phi$ mean surface or TOA short-wave radiation.
We included the word surface.
p.5 l22: Please, specify what the atmospheric forcing variables from the MAR model are
We did so.

Please add a table with model parameters and parameter values used in the main text and analysis.
We will do so, if this fits into the format (article or brief communication)

[Figure]

Fig. 2: add units to axis labels; duplicate y-axis labels ("PDD", "ETIM", and "dEBM")
We changed this.

Fig. 3: the min/max colors are really dark and hard to see
We tried to improve the colorbar.

./BILDER/Fig_S1.jpg

**Fig. AC1-4.** Monthly mean melt period temperature $T_{MP}$ and PDDs as functions of monthly mean near surface air temperature $T_a$. Crosses reflect monthly mean $T_{MP}$ as calculated from hourly near surface air temperature data of 18 PROMICE stations. Red and green points reflect PDD calculated from $T_a$ assuming a constant standard deviation of $3.5^oC$ and $5^oC$ respectively.

```
./BILDER/Fig_S2.jpg
```

**Fig. AC1-5.** Monthly mean wind speed during melt periods $u_{MP}$ as a function of monthly mean near surface air temperature $T_a$.

./BILDER/Fig_S3.jpg

**Supplement:**

[revised manuscript text omitted]

---

## Author Comment (AC2) · 10 Oct 2018

**1 General remarks**

We would like to thank all reviewers for the reading of the manuscript and for their thoughtful and very constructive comments. In particular addressing the sensitivity of the model and questioning the individual parameter choices has led to some interesting results. In the following, reviewer comments are printed in blue and manuscript excerpts are printed in grey. A texdiff-file is also provided for the revised manuscript. Also, I provide all figures from the Supplement at the end of this document. I will answer all points individually further below , but before that I would like to introduce some modifications of the dEBM, which were a side product of the revision process.

1.) I realized, that the minimum solar elevation angle was based on an incorrect estimate of the solar flux density at the surface ($\tau \widehat{S}_r = 600\,\mathrm{W\,m}^{-2}$.) Upon closer inspection I could identify the error in the calculation and corrected this to $\tau \widehat{S}_r = 800\,\mathrm{W\,m}^{-2}$, which results in a smaller minimum elevation angle of $\Phi = 17.5^o$.

2.) Analysing hourly PROMICE data, I found that $PDD_{\sigma=3.5}$ better represents melt period temperatures than $PDD_{\sigma=5}$ (approximated with a constant standard deviation of $\sigma = 3.5\,^\circ\mathrm{C}$ instead of $\sigma = 5\,^\circ\mathrm{C}$).

All experiments and analyses have been repeated, using $PDD_{\sigma=3.5}$ and minimum elevation angle $\Phi = 17.5^\circ$. We found that these corrections don't change the results qualitatively.

Furthermore we have included a section on the sensitivity of the scheme in the main manuscript. Depending on the editors decision this may as well be moved to the supplement:

**2 Sensitivity to model parameters and boundary conditions**

[revised manuscript text omitted]

- As pointed out by the editor, a distinction should be made when discussing a melt model alone (which is a somewhat ar-

25  tificial construct in isolation) versus an energy balance model, which may calculate many variables that are useful for ice-sheet modeling (ice temperature, albedo, refreezing, smb). Thus I see PDD, ETIMs, ITM (see next point) and the model presented here in a similar category – melt models that can be used as a subcomponent of smb models – while SEMIC and full EBMs are more wholistic solutions. A bit of clarity here on these definitions would improve the manuscript greatly.
We have included a more detailed description of SEMIC:

30  Introduction: Krapp et al. (2017) have formulated a complete surface mass balance model including accumulation, surface melt and refreezing (SEMIC) which can be used with daily or monthly forcing. SEMIC predicts the surface mass balance with a daily time step, but implicitly accounts for the sub-daily temperature variability in the surface layer of the ice to account for diurnal freeze-melt cycles.

35

-"ETIM" as used throughout the text seems to be the wrong term for the comparison being made here. ETIMs involve a "temperature index" such as PDD (Hock et al., 2003; Pellicciotti et al., 2005). This is why, in Robinson et al., 2010, we opted to create the name ITM for the model of Eq. 14, since we saw it simply as an "insolation-temperature melt model".
- Equation 14 is not correct, in representing the approach of Pollard (1980) or Robinson et al. (2010). The second term should

40  not contain PDDs, but the mean near-surface air temperature relative to the melting point: $k_1 * (T_a - T_0)$
Admittedly, this was formulated too sloppy. Trying to correctly incorporate the ITM concept, I noticed that this method has problems to reproduce melt rates at low temperatures. In the context of a full SMB-model this would propably carry no weight as refreeze is usually balancing the surface melt at low temperatures. However, the calibration to MAR melt rates is hampered by the biased low temperature melt rates. In consequence, our calibration would yield parameters, which fail to skillfully

45  reproduce the surface melt rates at warm temperatures. I have decided to stick to a scheme which incorporates PDD, but we now refer to this scheme as $dEBM_{const}$. Comparing to a dEBM version with constant parameters also meets your suggestion

further below. We have reformulated the first part of section

**First evaluation of the scheme:** The dEBM and two empirical schemes are calibrated and evaluated using the state-of-the-art regional climate and snow pack model MAR (Fettweis et al., 2017) as a reference.

The elevation angle used in the dEBM is estimated as $\Phi = 17.5°$, aplying Eq. (9) with a typical albedo of 0.7 and $\tau \widehat{S}_r = 800\,\mathrm{W\,m}^{-2}$ being roughly estimated from the summer insolation in the ablation regions (Eq. **??**). This estimate corresponds to a transmissivity of $\tau \approx 0.6$ which is in good agreement with Ettema et al. (2010). Further, the dEBM is optimized to reproduce the total annual Greenland surface melt averaged over the entire MAR-simulation by calibrating the background melting condition as $\overline{T}_a > -6.5\,°\mathrm{C}$ and the parameter $\beta = 10\,\mathrm{W\,m}^{-2}\mathrm{K}^{-1}$. We then apply the scheme to $\overline{SW}_0$, $PDD_{\sigma=3.5}(\overline{T}_a)$ and albedo $A$ from a MAR-simulation of Greenland's climate (years 1948 to 2016) (Fettweis et al., 2017) and compare estimated melt rates with the respective MAR melt rates.

Two empirical schemes are considered in the same way: a PDD-scheme based on $PDD_{\sigma=5}(\overline{T}_a)$, as defined and calibrated in Krebs-Kanzow et al. (2018) and a scheme, in the following refered to as $dEBM_{const}$, which is a simplified variant of the dEBM where parameters are constant in time and space:

$$M = ((1 - A)\overline{SW}_0 + k_1 PDD_{\sigma=3.5}(\overline{T}_a) + k_2)\frac{1}{\rho L_f} \qquad (1)$$

with $k_1 = 10\,\mathrm{W\,m}^{-2}\mathrm{K}^{-1}$ and $k_2 = -55\,\mathrm{W\,m}^{-2}$. The $dEBM_{const}$ is very similar to the ITM-scheme and also uses similar parameters as in Robinson et al. (2010), but includes PDD instead of temperature, which particularly yields different results for low temperatures. As in Robinson et al. (2010), we treat $k_2$ as a tuning parameter to optimize the scheme and also use $\overline{T}_a > -6.5\,°\mathrm{C}$ as a background melting condition.

The computational cost of the dEBM in this application is very similar to the other two schemes as parameters are computed only once prior to the application. All schemes reproduce the total annual Greenland surface melt averaged over the entire MAR-simulation of $489\,\mathrm{Gt}$ with a relative bias not exceeding $1\%$ (the mean bias is $0.4\,\mathrm{Gt}$ for the PDD scheme, $-0.6\,\mathrm{Gt}$ for the $dEBM_{const}$ and $-2.0\,\mathrm{Gt}$ for the dEBM). These calibrations are primarily conducted to facilitate a fair comparison between the different schemes and are not necessarily optimal for other applications.

- SEMIC also supports the input of monthly temperature data, although the model itself is calculated on daily time steps (from Krapp et al., 2017: "In principle, the use of monthly input data is also supported but would require interpolation to daily time steps."). I would additionally note that SEMIC is open source and prepared to run easily with MAR data as input, making its comparison with dEBM feasible if the authors wanted to be more thorough. It would certainly be convincing if it could be explicitly shown that dEBM can do a better job than SEMIC for a much lower cost. (This point is only a suggestion, and I would not consider it necessary for revision.)

I now discuss SEMIC more specificly in the Introduction (see above). I would also be very interested in this comparison. The intention behind the comparison of the different schemes in the presented paper, however, is not a complete intercomparison, but rather to demonstrate, that the latitudinal bias found in other schemes, might be reduced, if we account for the diurnal cycle of radiation.

I find the approach outlined here quite elegant and the physical derivation is nicely described. However, then I am surprised to see PDD pop up in Eq. 6 again. Would it not be simpler keep $(T_a - T_0)$ here? The only reason to use PDDs is to incorporate a measure of variability in $T_a$. But it seems to me that if you want to include the variability around $T_a$ in the melt model, it would be more appropriate to apply it to the whole equation rather than just to the temperature term (ie, calculate the average melt rate from the distribution of melt rates for the month).

We have added a few lines to

**The daily melt period and its energy balance:** Near surface air temperature measurements from PROMICE stations on the GrIS reveal a good agreement between monthly mean temperatures of the daily melt periods and the $PDD_{\sigma=3.5}$ approximated as in Braithwaite (1985) from monthly mean near surface temperature $\overline{T}_a$ and a constant standard deviation of $\sigma = 3.5\,°\mathrm{C}$ (Fig. S1 in the supplement).

And added the following to the

**Supplement: Mean surface temperture and wind speed of melt periods from observations** $T_a$ is the monthly mean temperature and thus also includes temperatures outside of the daily melt period. The strategy in our paper is to only consider that part of the day, when the ice is warm enough to melt. We thus need to estimate the mean temperature during this melt period. To illuminate the relation between $T_a$ and $T_{MP}$, we analyzed hourly climate data from PROMICE (Ahlstrom et al., 2008) weather stations: 2m air temperature $T_a$, surface temperature $T_{surf}$, albedo A and short wave radiation SW. In analogy to the dEBM, we determine the melt period for each month by identifying those hours which comply with the conditions

$$\overline{(1-A)SW} > 71.9 Wm^{-2}$$

and

$$\overline{T_{surf}} > -0.01^o C$$

. The bars denotes hourly data taken from the monthly mean diurnal cycle. We analyzed 18 PROMICE stations which cover a period of up to ten years (2008-2017) and identified 390 monthly mean diurnal cycles which exhibit a melt period acording to our above definition. We don't need to resort to a minimum elevation angle here, as hourly radiation is available. Likewise the background melting condition is replaced by the condition, that hourly surface temperature data must be near melting point. Indeed, the PROMICE data indicate that PDD is quite a good proxy for the monthly mean temperature of the melt period $T_{MP}$. Using a constant standard deviation of $3.5\,^{\circ}$C exhibits a particularly good fit (Fig. S1). Furthermore analyzing the mean wind speed during the above melt periods, we find on average a wind speed of $u_{MP} = 3.8\,\mathrm{m\,s^{-1}}$ (Fig. S2).

Why PDD is such a good estimate of $T_{MP}$ is not completely clear. It would be interesting to develop alternative estimates for $T_{MP}$.

Looking into the observational data, we also noticed that daily melt periods are considerably longer than in our original MAR-based study (up to 20 hours). In fact this is how I realized that the estimated minimum elevation angle must be incorrect. The newly estimated $\Phi = 17.5^o$ yields longer melt periods and generally agrees with the PROMICE data.

If you follow the path above, this change would make Eq. 6 essentially equivalent to Eq. 14 (also without PDDs), and it maintains its physically-based origins and makes it obvious that the key differences are: - The term $q_\Phi$, which scales the insolation according to the time it is actually available. - The term $dt_\Phi$, which scales the melt according to the time when it is relevant. - The derivation of the constants $c_1$ and $c_2$. I note that the used values of $c_1 = 13.5$ and $c_2 = -71.9$ are not too far from values used in Eq. 14 for $k_1 = 10$ and $k_2 = -60.5$. It would be interesting to understand if this is systematic, that generally $c_1 > k_1$ and $k_2 > c_2$, to compensate for the lack of $q_\Phi$ and $dt_\Phi$ terms. For example, if in Eq. 6, you set $q_\Phi = 1$ and $dt_\Phi = dt$, how well does your model perform (after retuning the constants) – as well as before, or is the performance degraded? In other words, I would be happy to see an analysis that specifically shows the value of incorporating the diurnal terms to the model

I think with introducing $dEBM_{const}$, this suggestion is implemented now. $dt_\Phi < dt$ in most places and accordingly $c_1 > k_1$. Understanding $q_\Phi$ is more complicated. We have reformulated the part about the radiative contribution and hope this is more clear, now.

**First evaluation of the scheme:** The radiative contribution in the dEBM becomes less efficient for long melt periods, as the same insolation must balance the outgoing longwave radiation for a longer time. On the other hand, radiative contribution can also decrease towards short melt periods, if the sun only marginally rises above the minimum elevation angle at solar noon. This effect becomes important for higher estimates of the minimum elevation angles in high latitudes (Sect. 4).

**– Minor comments —–**

We would like to comment on the following minor comments
: Page 2, line 1: "Another empirical aproach, the enhanced temperature-index method, ETIM" <= In addition to the fact that I believe ETIM is the wrong term here, as I already mentioned, ETIM refers to a class of models that can take many forms that generally extend PDD in various ways, not to a specific model formulation. Therefore, I would rephrase here. Alternatively, you can use the term "ITM", which does refer to the formu- lation of Pollard (1980). Or, a more descriptive term for this model

would be "linearized EBM" (Pollard, 1980).
We now use $dEBM_{const}$

Page 3, line 5 (Eq. 3): I see no reason why $e_i$ should appear multiplied with $LW_{down}$. This is only relevant for $LW_{up}$ (as
5   in Eq. 4), correct?
To my understanding, $e_i$ is also influencing, how much LW radiation is absorbed, as a good emitter is also a good absorber.

Page 4, line 23: What is the calculation of $\Phi = 23.5$ used for later? As I understand all tests were using MAR albedo, etc.
Is this just an example?
10   Actually $\Phi$ is a crucial parameter, as $\delta t_\Phi$ and $q_\Pi$ will change with $\Phi$.

Page 5, line 11: "Eqations (6) and (14) appear formally similar, with the first and third term representing the radiative con-
tribution and the second term representing the PDD contribution." <= This sentence is contaminated by the mistake in Eq. 14,
15   however, just thinking about it in terms of Eq. 6, it is clear from the derivation that the first term represents shortwave radiation
and the second and third terms represent the net longwave radiation and heat fluxes from R combined. Please rephrase.
We rephrased:
Equations (6) and (14) appear formally similar, with the second term being temperature dependent (the "temperature contri-
bution") and the first and third term being independent of temperature and only depending on solar radiation (the "radiative
20   contribution").

Page 5, line 11-21: Generally, I find this paragraph difficult to follow. Is the "flat elliptic" referring to the orbital configuration
of the Earth, or some pattern in the figure itself? Does "going along with" mean "causing"? I find that "PDD contribution" a not
very convenient name for the second term in Eq. 6, since it is easily confused with the PDD melt model itself in this context. I
25   would consider serious revision here for clarity.
We reformulated this paragraph (ecliptic was the wrong term):
Fig. 1a illustrates the radiative and Fig. 1b the temperature contributions as diagnosed from the MAR simulation in compar-
ison to the respective contribution from the $dEBM_{const}$. On the GrIS the radiative contribution can exceed $25\,\mathrm{mm\,d^{-1}}$ in
the summer months and the two schemes appear qualitatively similar. The radiative contribution in the dEBM becomes less
30   efficient for long melt periods, as the same insolation must balance the outgoing longwave radiation for a longer time. On
the other hand, radiative contribution can also decrease towards short melt periods, if the sun only marginally rises above the
minimum elevation angle at solar noon. This effect becomes important for higher estimates of the minimum elevation angles
in high latitudes (Sect. 4). The temperature contribution of the dEBM does not exceed $15\,\mathrm{mm\,d^{-1}}$ (Fig. 1b) and becomes more
efficient with longer melt periods and would agree with the $dEBM_{const}$ for a melt period of 18 hours.
35
Page 5, line 25: "defective input" <= I'm not quite sure what you want to say with this sentence, consider rephrasing somewhat.
Wouldn't it be possible to make your ideal input data "defective" for testing purposes, if that was your goal?
We slightly rephrased:
With respect to error propagation the PDD-scheme might be more robust and , as it only requires temperature as a forcing and
40   only distinguishes between snow and ice but does not require albedo.

t does not seem appropriate to limit the comparison of dEBM to points that satisfy $T_a > -6.5C$. Either the value of $T_{min} =$
$-6.5C$ is adequate, or $T_{min}$ should be set to a lower value. In either case, the correct choice of this threshold should be re-
flected by the comparison to MAR melt. Based on the horizontal line of dark blue points in Figs. 1 and 2, I have to guess that
45   the threshold chosen here is too high, or for some reason the dEBM underestimates melt at low temperatures. This should be
discussed in the paper clearly. The horizontal line of dark blue points in Fig. 2 was related to the incorrect estimate of minimum
elevation angle. We removed the RMS part from the main text and included the root mean square errors of the mean 1948-2016
local yearly surface melt rates in Fig. 3. We did not use the background melting condition for this calculation. The idea to use
$T_{min}$ arose as I wanted to limit the analysis to the ablation region and melt season. I see now that this can bias the statistics (it

did not substantially, though)

Page 8, line 13: "This threshold temperature should be considered as a tuning pa- rameter" <= I had understood this $T_{min}$ simply to be a cost-saving measure, to avoid calculating the melt model for points where melt would be zero. However, this sentence makes me believe that the parameter is more important than I realized. Please elaborate on the role of $T_{min}$ more in the derivation section for clarity

This is now discussed in the sensitivity section (Sect. 4).

We fully followed the following minor comments and corrected the manuscript accordingly:

Units and variables: Please check the units carefully. For example, $T_a$ is in Kelvin, but then $T_{min} = (T_0 - 6.5)K$, right? Also, in Eq. 14, is the first term "$SW_0$" the same as "$SWD_0$" defined earlier in the text? Please keep the same terms throughout
. Page 1, line 14: information on => information about
Page 1, line 15: refreeze => refreezing
Page 1, line 23: computational => computationally
Page 1, line 24: temperatures. => temperatures as input.
Page 1, line 25: or paleo-temperature => and paleo-temperature
Page 2, line 1: aproach => approach
Page 2, line 19: a surface melt rate => a non-zero/positive surface melt rate
Page 3, line 7: Per definitionem => By definition
Page 3, line 23 (Eq. 7): It looks like $c_1$ is missing the term $e_a$, following the current equation formulations.
Page 4, Section 2.1: Please make sure to use the same variables and notation as in the rest of the text. I guess that the elevation angle $\Phi$ in the previous section is the same as the elevation angle $\Theta$ in Sect. 2.1.
Page 4, Eq. 13: I would suggest adding the intermediate definition of $q_\Phi$ here to remind readers of your previous definition: $q_\Phi = SW_\Phi/SW_0 =$ [full definition]. Also again be clear about SW versus SWD.
Page 5, line 11: Eqations => Equations
Page 5, line 23: derived => obtained?
Page 5, line 27: due to => Given
Page 7, line 1: biasses => biases
Page 7, line 6: refreeze => refreezing
Page 7, line 6: used together with the enhanced temperature index method in =>presented by
Page 8, line 16: Depending on application => Depending on the application

**4   Response to second referee (anonymous)**

Page 4 line 12 TOA is introduced for "top of the atmosphere" but only used once on line 15 then not used on line 16 (where there are hyphens between the words). Is an initialism really needed?
It is now used more often

Page 4 line 21 "Choosing $\beta = 10$ ...". If that is a choice, i.e. if alternative values could have been chosen, then the reason for this specific choice should be given, e.g. cited or explained. If however it is the only reasonable value then it's not a choice and "using "would be better than "choosing"
We now make clear that this is the outcome of an calibration: Further, the dEBM is optimized to reproduce the total annual Greenland surface melt averaged over the entire MAR-simulation by calibrating the background melting condition as $\overline{T_a} > -6.5\,\mathrm{K}$ and the parameter $\beta = 10\,\mathrm{W\,m^{-2}\,K^{-1}}$

Page 4, bottom, section 3 Just a general comment that any further citations or justifications for the values of coefficients used that can be included would be useful

We now reference **?** and Orvig (1954) which agrees well with our independently calibrated parameters.

In the introduction it is mentioned that the PDD scheme is computationally inexpensive (page 1 line 23) and that energy balance models could have their computational costs educed (page 2 line 7) but the evaluation makes no mention of the computational costs of dEBM and the other schemes. I think it would be useful to include a brief comment on the relative computational costs in section 3.

We added the following line to section 3: The computational cost of the dEBM in this application is very similar to the other two schemes as parameters are computed only once prior to the application.

The following suggestions are obsolte after the modification of the manuscript: Page 5 line 16 "going along with .... " would "corresponding to..." read better?
Page 5 line 30 This refers to the blue points in Fig 2 panel 3 at 0 on the y-axis. I think this should be stated in the text.
Page 8 line 16 This is a new paragraph, should it be

We fully followed the following recommendations: Page 1 line 23 "computational" should be "computationally"
Page 2 line 27 "Further, we define the ratio between ....." would "$q_\Phi$ is the ratio between..." read better?
Page 2 line 27 and 28 "SW" is used for the mean solar radiation. I assume the S and W stand for Short Wave, so it would be better to state that here
Page 3 lines 24 and 26 and Page 4 line 10 These lines appear to have been indented /tabbed
Page 4 line 22 "-6.5K" $^oC$ not K

Page 5 line 11 Misspelling "eqations"
Page 5 line 15 "GrIS" is used without definition. Whilst it is a well known abbreviation, especially for this journal, it maybe better to define
Figure S1, caption "meltrates" should be 2 words. Misspelling "lenght". Refers to PDD, ETIM and DEBM as a), b) and c) but they are not labelled as such in the figures. Also "Identity... black line" is not shown (I think perhaps the caption is for an earlier version of the figure?)

**5   Response to third referee (Mario Krapp)**

**Major comments** - Solar elevation angle and surface slope: Whereas large parts of the Greenland ice sheet are rather flat its margins, where most of the melt occurs) are not and glaciers are even more sensitive to the slope of the embedding terrain. I suspect that the daily solar elevation angle depends on how the ice surface faces the sun. How much of an effect would a surface slope have and could that be included in Sect. 2.1?
Indeed, the melt period may be extended/shortened by a southward/northward slope. If we would want to account for this, it would make it necessary to perform a projection of the solar radiation to the surface before estimating the minimum elevation angle locally. The slopes on the 20km grid of the MAR-simulation rarely exceed $1\%$, which could change the minimum elevation angles by $\approx 1^o$. Seeing that the manuscript is already quite lengthy, I decided to not include this into the manuscript.

- I expect the atmospheric transmissivity (Sect. 2.1) to decrease with increasing solar zenith angle. How much of an effect would that have?
This is a very good point, I would estimate that this effect may increase the elevation angle by up to $2^o$ relative to an estimate using a constant transmissivity. It is difficult to account for this effect in an objective way, but luckily the scheme is quite insensitive to minor changes in the minimum elevation angle and it appears sufficient to only do a rough estimate. I added the

following sentence to the Sect. 4:

As one can expect, that transmissivity decreases towards the morning and afternoon hours, it may be justified to reduce the estimate of $\tau \widehat{S_r}$ by a few percent.

I think that using a single parameter for the emissivity of air ( $\epsilon_a$ ) is also too simplistic and the contribution of cloud cover is missing. $LW_{down}$ is parameterised using $\epsilon_a$ , which is the clear sky emissivity but how do you deal with cloudy skies? In fact, $\epsilon_a$ can vary between 0.7 (clear sky) and 1.0 (fully overcast). Therefore, the value for $c_2$ can vary between -90 and 0 W/m2 if you account for varying $\epsilon_a$. That means that a full overcast sky would add about 90 W/m2 to the surface energy uptake Q. A very valid point, I did not consider this originally. The dEBM concept propably comes to its limits here. I have added this point to the sensitivity section as given above. However for continental ice sheets (i.e. Greenland and Antarctica and, in cold climates, the North American and Fennoscandian ice sheets) the clear sky assumption appears justified.

I think in Eq. (7), $\epsilon_a$ is missing in the term for $c_1$ , ... If that is the case $c_1$ also yields a different value in line 25 on page 4 and my above argument about varying $\epsilon_a$ implies that $c_1$ can vary 13 and 14.4 W/m2K This is an error which was only in the text and not in my dEBM function. I corrected the text accordingly.

Sensitivity of model parameters:... We now include the section about sensitivity, as stated above

The PDD component of dEBM is in general smaller than in ETIM (Fig. 1b). Obviously, the PDD contribution of dEBM would be larger for a larger $\beta$ which can range between 7 to 20 W/m2K as you said earlier This has changed after correcting and modifying various details. Also citing **?** we now provide better constraints for the choice of $\beta$.

I would like to see a plot showing the time series of monthly melt and different diagnostics (as is shown in the supplement). For example, melt rates and its individual components (the PDD and the ETIM-related term) in Eq. (6), or the parameterised short- and longwave radiation SW and $LW_{down}$ would help the reader to understand what the model is doing internally. Specifically it would be nice to see how $q_\Phi$ , which is the novel part of your melt scheme, changes over time. Primarily, $q_\Phi$ and $\delta_\Phi$ affect surface melt latitudinally, and to some degree seasonally. Perhaps, the effects are sufficiently illustrated in the new Fig. 4. I am hesitant to add another figure on the seasonal effect, as the paper seems already quite long.

To me everything in the conclusion, except for the first paragraph, is more like a "summary and discussion" section than an actual conclusion. Please revise. We changed the title of the section accordingly.

I guess if you consider a revision as article you can easily move Figure S1 (which is the only item in the supplement) to the main text This can be easily done, but I would leave this decision to the editor.

Out of curiosity (not needed for the revison): If the melt scheme just uses a few input parameters, is it possible to force it with atmospheric data from available observations of the GrIS? For example, GC-MET (http://cires1.colorado.edu/steffen/ At least the PROMICE data have a high frequency, so that better estimates should be possible, if a full energy balance model is used. Nevertheless, I can imagine that the scheme could be modified in a way, so that distributed melt estimates could be derived from satelite data in combination with weather station data. Also it could be possible to estimate melt rates from glaciers where weather stations only exist below the glacier. In both cases I would think that the scheme would have to undergo considerable modification. I would be indeed interested to discuss this with people from the observational community.

**Minor comments**

p2 ll.27-29: It is not clear whether $SW_0$ or $SW_\Phi$ mean surface or TOA shortwave radiation.

We included the word surface.

p.5 l22: Please, specify what the atmospheric forcing variables from the MAR model are

We did so.

5  Please add a table with model parameters and parameter values used in the main text and analysis.

We will do so, if this fits into the format (article or brief communication)

Fig. 2: add units to axis labels; duplicate y-axis labels ("PDD", "ETIM", and "dEBM")

We changed this.

Fig. 3: the min/max colors are really dark and hard to see

We tried to improve the colorbar.

[revised manuscript text omitted]

van den Berg, J., van de Wal, R., and Oerlemans, H.: A mass balance model for the Eurasian ice sheet for the last 120,000 years, Global and Planetary Change, 61, 194–208, https://doi.org/10.1016/j.gloplacha.2007.08.015, 2008.

[Figure]

**Figure 4.** Sensitivity of the dEBM: June surface melt rate as predicted for $SW_0 = 200 \, \mathrm{W \, m^{-2}}$, A = 0.7, $T_a = -3 \, ^\circ\mathrm{C}$ (left curves) and $T_a = 3 \, ^\circ\mathrm{C}$ (right curves). Black: predictions with parameters as used for the presented simulation of Greenland's surface melt. Green: parameters are recalculated using the minimum (solid) and maximum (dashed) obliquity of the last 1 million years. Blue: parameters are recalculated after the minimum elevation angle is adjusted to a reduced solar density flux at the surface of $\tau \widehat{S}_r = 700 \, \mathrm{W \, m^{-2}}$ (solid), $\tau \widehat{S}_r = 600 \, \mathrm{W \, m^{-2}}$ (dashed), $\tau \widehat{S}_r = 500 \, \mathrm{W \, m^{-2}}$ (dots). Red: parameters are recalculated after the minimum elevation angle is adjusted to an intensified solar density flux at the surface of $\tau \widehat{S}_r = 1150 \, \mathrm{W \, m^{-2}}$. The $dEBM_{const}$ predicts 0 mm/day for $SW_0 = 200 \, \mathrm{W \, m^{-2}}$, A = 0.7, $T_a = -3 \, ^\circ\mathrm{C}$ and 9 mm/day for $SW_0 = 200 \, \mathrm{W \, m^{-2}}$, A = 0.7, $T_a = 3 \, ^\circ\mathrm{C}$ (black dots).

[Figure]

**Figure S-5.** Monthly mean melt period temperature $T_{MP}$ and PDDs as functions of monthly mean near surface air temperature $T_a$. Crosses reflect monthly mean $T_{MP}$ as calculated from hourly near surface air temperature data of 18 PROMICE stations. Red and green points reflect PDD calculated from $T_a$ assuming a constant standard deviation of $3.5^oC$ and $5^oC$ respectively.

[Figure]

**Figure S-6.** Monthly mean wind speed during melt periods $u_{MP}$ as a function of monthly mean near surface air temperature $T_a$.

[Figure]

**Figure S-7.** Upper panel: total yearly surface melt of the years 1948-2016 from MAR (black) and as predicted a) Total Greenland surface melt from 1948 to 2016 as simulated by MAR (black) and predicted from PDD-scheme (blue), $dEBM_{CONST}$ (green) and dEBM (red). Lower panel: yearly bias of total yearly surface melt predicted by PDD-scheme (blue), $dEBM_{CONST}$ (green) and dEBM (red) for the 1948–2016 period relative to MAR.